# Common Event Tethering to Improve Prediction of Rare Clinical Events

Quinn Lanners*[1]        Qin Weng*[1]        Marie-Louise Meng[2]        Matthew M. Engelhard[1]

[1]Department of Biostatistics & Bioinformatics, Duke University School of Medicine, Durham, NC, USA
[2]Department of Anesthesiology, Duke University School of Medicine, Durham, NC, USA

## Abstract

Learning to predict rare medical events is difficult due to the inherent lack of signal in highly imbalanced datasets. Yet, oftentimes we also have access to surrogate or related outcomes that we believe share etiology or underlying risk factors with the event of interest. In this work, we propose the use of two variants of a well-known approach, regularized multi-label learning (MLL), that we hypothesize are uniquely suited to leverage this similarity and improve model performance in rare event settings. Whereas most analyses of MLL emphasize improved performance across all event types, our analyses quantify benefits to rare event prediction offered by our approach when a more common, related event is available to enhance learning. We begin by deriving asymptotic properties and providing theoretical insight into the convergence rates of our proposed estimators. We then provide simulation results highlighting how characteristics of the data generating process, including the event similarity and event rate, affect our proposed models' performance. We conclude by showing real-world benefit of our approach in two clinical settings: prediction of rare cardiovascular morbidities in the setting of preeclampsia; and early prediction of autism from the electronic health record.

## 1 INTRODUCTION

Predicting the risk of an adverse event is one of the most common applications of machine learning in medicine. The ability to identify patients at risk of developing a disease, suffering a side-effect from a treatment, or seeing a drastic change in their health allows clinicians to take the appropriate steps to prevent or mitigate the negative outcome.

However, many clinically important events are highly impactful but extremely rare, resulting in imbalanced training datasets and making it difficult to train an effective model [Megahed et al., 2021, He and Garcia, 2009]. Yet, oftentimes we also have access to surrogate or related outcomes that are more common, yet share etiology or risk factors with the rare event(s) of interest. In such scenarios, we would like to leverage information from the more common event(s) to help us predict the rare event(s).

In this paper, we consider two separate medical contexts in which rare event prediction is challenging, but related outcomes are available to enhance learning. Our first task aims to predict stroke in patients with hypertensive disorders of pregnancy (HDP). HDP put pregnant people at high risk for stroke as well as several other severe complications, such as hypertensive crisis, that are more common [Meng et al., 2023]. Our second task aims to predict autism at an early age based on electronic health record (EHR) data collected from birth through age 18 months. While autism diagnosis is uncommon, the condition shares clinical features and risk factors with several more common neurodevelopmental conditions, including ADHD and developmental delays. In both of these tasks, the events of interest are rare and we would like to share information across related outcomes to help improve model performance.

In such settings, it is natural to consider multi-label learning (MLL) as a way to share information between outcomes. Indeed, the concept of MLL has been studied extensively and successfully applied to many prediction tasks in medicine [Zhang et al., 2015, Li et al., 2015, Zufferey et al., 2015, Ge et al., 2020]. However, typical MLL methods aim to improve performance across all event types and were not designed specifically to improve rare event prediction by leveraging related but more common events. Moreover, we know very little, both empirically and theoretically, about the conditions under which MLL confers benefit for rare event prediction tasks. For the same reasons, we do not know which MLL methods are well suited to the clinical scenarios previously described.

---

*Equal contribution.

In this work, we expand upon existing MLL literature to focus on rare event prediction. Specifically, we provide insight into how event rarity and underlying similarity affect MLL performance. Motivated by this insight, we propose a variant of *regularized* MLL when working with rare events. Our approach bridges early MLL literature, which largely focused on task-sharing shrinkage/priors, with more recent work, which has focused on representation learning-based methods [Evgeniou and Pontil, 2004, Zhou et al., 2012, Huang et al., 2019, Zhu et al., 2021]. We show that a combination of these two may be suitable for cases where clinical events share latent risk factors but the events we are trying to model are extremely rare.

**Contributions.** Our work contributes to the field of MLL learning and rare event prediction with a focus on its applications to medical settings. Our main contributions are as follows:

- Propose a variant of regularized MLL, which we call common event tethering (*CET*), that is specifically suited for rare events.

- Provide theoretical analysis identifying conditions on event similarity under which CET is superior to standard shrinkage estimators.

- Analyze the effect of event similarity and event rate on the effectiveness of CET for rare events both theoretically (Sections 4) and via simulation (Section 5).

- Demonstrate the benefits of our approach when predicting rare cardiovascular morbidities in pregnant people with HDP and predicting autism likelihood in early childhood.

This paper proceeds as follows. In Section 2, we outline the setup of our problem statement and discuss related work. We proceed to introduce our MLL methods, which we call *common event tethering logistic regression (CET-LR)* and *common event tethering neural network (CET-NN)*, in Section 3. We then provide theoretical results on the asymptotic properties of our estimators in Section 4. Section 5 uses simulation to highlight how the underlying similarity between and rate of events affects the performance of our CET methods. We ultimately highlight the benefits of these methods in our two real-world applications in Section 6.

## 2 SETUP AND RELATED WORK

We consider the setting where we have a dataset $\mathcal{D}_n = \{(\mathbf{x}_i, \mathbf{y}_i)\}_{i=1}^n$ of $n$ independent samples where, for each patient $i$, $\mathbf{x}_i$ is a $p$-dimensional feature vector and $\mathbf{y}_i \in \{0, 1\}^M$ is a vector of $M$ binary outcomes. We consider the case where $M = 2$, though our results can easily be generalized to settings where $M > 2$. Without loss of generality, we let $y_{i,1}$ be the label for a rare event of interest such that the event rate, $\frac{1}{n} \sum_{i=1}^n y_{i,1}$, is low (e.g. 0.01, 0.001, ...) and

we let $y_{i,2}$ be the label for another more common, related event.

We assume that the probabilities of $y_{i,1}$ and $y_{i,2}$ can be written as

$$
\begin{aligned}
P(y_{i,1} = 1 | \mathbf{x}_i) &= \sigma(\boldsymbol{\theta}_1' h(\mathbf{x}_i)), \text{ and} \\
P(y_{i,2} = 1 | \mathbf{x}_i) &= \sigma(\boldsymbol{\theta}_2' h(\mathbf{x}_i)).
\end{aligned}
\tag{1}
$$

In Equation 1, $\mathbf{x}_i$ is assumed to include a constant feature, $\sigma$ is the sigmoid function, $\boldsymbol{\theta}_1, \boldsymbol{\theta}_2 \in \mathbb{R}^d$, and $h : \mathbb{R}^p \to \mathbb{R}^d$ is a function that maps $\mathbf{x}_i$ to a $d$-dimensional vector of latent features.

Note that when $h(\mathbf{x}_i) = \mathbf{x}_i$ this simplifies to a standard logistic model. In this way, we view the probabilities of $y_{i,1}$ and $y_{i,2}$ as being determined by two separate logistic models on either the input features themselves or some latent features that we can learn. $\boldsymbol{\theta}_1$ and $\boldsymbol{\theta}_2$ then act as the $d$ dimensional parameter vectors for the corresponding outcome.

### 2.1 LOGISTIC REGRESSION & L2 REGULARIZATION

Logistic regression (LR) is a commonly employed technique for classification. However, LR is infamous for being biased and unstable in small sample sizes [Firth, 1993]. In the context of rare events, Wang [2020] showed that the convergence rate of the LR MLE is much slower; specifically $O_p(n_1^{-\frac{1}{2}})$ where $n_1 = \sum_{i=1}^n y_{i,1}$.[*] In turn, the amount of information available for rare event problems is directly related to the number of events in the dataset. This essentially decreases the effective size of a dataset, making LR for rare events particularly unstable and susceptible to problems with small sample sizes. To offset this unstable behavior, ridge regression was proposed by Hoerl and Kennard [1970] and later extended to LR by Cessie and Houwelingen [1992]. This ubiquitous technique places an $L_2$ penalty on model coefficients to decrease the variance of the estimate at the expense of a higher bias.

In the setting of rare events, where little information is present in the data, such regularization can lead to better out-of-sample performance [Pavlou et al., 2016]. However, as noted by Šinkovec et al. [2021], tuning of the penalty parameter can be unstable and ultimately penalization alone cannot overcome insufficient sample sizes or extreme class imbalance [Riley et al., 2020, Blagus and Lusa, 2010].

### 2.2 MULTI-LABEL AND MULTI-TASK LEARNING

One approach to combat the small effective sample size of rare events is to leverage information from related events. MLL methods are designed to predict a set of outcomes from

---

[*]The standard convergence rate is $O_p(n^{-\frac{1}{2}})$ [Wang, 2020].

a collection of input features, often sharing information between labels [Aly, 2005, Zhang and Zhou, 2013, Liu et al., 2021]. Related to MLL methods, multi-task learning (MTL) methods can leverage information from different tasks trained on different datasets to improve performance across all tasks [Zhang and Yang, 2021]. In general, multi-label learning can be seen as a form of multi-task learning where the same dataset is used to learn about each task.

We combine a specific form of regularization based information sharing with representation learning based information sharing to improve performance of rare event modeling. The idea of regularized MLL (or MTL) has been widely studied [Cao et al., 2019]. Evgeniou and Pontil [2004] and Evgeniou et al. [2005] were among the first to explore the topic in the context of kernel estimators. Zhou et al. [2012] used a version of fused-lasso with an $L_1$ penalty between the coefficient vectors and more recent works like He et al. [2019], Yu et al. [2020], and Alesiani et al. [2021] have imposed a variety of related penalties on coefficient vectors for scalable and interpretable MTL. More recently, Janati et al. [2019] used the Wasserstein distance for sparse regression, Tang et al. [2020] utilized evolutionary algorithms, and Bai and Zhao [2022] looked at regularization in multi-task deep learning problems.

The growth of deep learning methods has led to representation learning being used extensively for MLL [Huang et al., 2019, Liu et al., 2021]. Recent work has combined this approach with regularization based information sharing techniques by using methods such as manifold regularization [Zhu et al., 2021], full-order label correlation [Chen and Zhang, 2019], shrinkage methods [Han et al., 2010], and dimensionality-reduction [Huang et al., 2020].

Regularized MLL/MTL has become increasingly popular for medical applications [Hossain et al., 2021, Zhu et al., 2022], but limited work has focused on rare event prediction. Zhang et al. [2015] proposed a regularized MLL approach for the prediction of drug side effects, using regularization to inform feature selection for an ensemble model. Faletto and Bien [2023] considered a regularized ordinal regression method inspired by fused LASSO and designed to shrink towards proportional odds in settings where the rare event can be characterized as the most extreme of a set of ordered outcomes. Most similar to our approach is Lapedriza et al. [2007] which imposes a penalty between the coefficient vectors of logistic regression models for related tasks. Our approach builds on this in two important ways. It is the first to explore, via both simulation and theory, the impact of event relatedness and event rate when using shrinkage penalties like those proposed by Lapedriza et al. [2007] and our method. Secondly, we incorporate feature learning via a neural network architecture to allow our approach to extend to more complicated non-linear setups.

## 2.3 OTHER APPROACHES

There are various other approaches that are less related to our proposed method but can also be used for rare event prediction tasks. These approaches often employ similar regularization or information-sharing strategies but also include pre-processing steps and different machine learning model architectures.

Transfer learning is a well-known information sharing approach that aims to adapt a model trained on one task – typically with plentiful data – to a second task for which less data are available [Zhuang et al., 2020]. Unlike MLL and MTL, transfer learning involves training on these tasks in sequence rather than simultaneously. We note that multi-label, multi-task, and transfer learning are not mutually exclusive. For example, transfer learning can be used to train a multi-class classifier on new tasks.

There are also a number of single-label learning approaches that have been employed for rare event prediction. Firth logistic regression is a widely-used method for prediction of imbalanced binary outcomes [Puhr et al., 2017]. It introduces a penalization term which helps to eliminate bias in parameter estimation when dealing with rare events [Olmuş et al., 2022]. Ensemble-based machine learning algorithms, including gradient boosting and random forests, are also commonly used for rare event prediction due to their ability to model complex and non-linear relationships while also mitigating overfitting by drawing random subsamples during training [Shyalika et al., 2023].

Another traditional method to address data imbalance is using resampling algorithms, such as under-sampling (down-sampling) and over-sampling (up-sampling) [Barandela et al., 2004]. Under-sampling removes examples from the majority class, while over-sampling replicates minority class samples to balance the dataset. However, the former may lead to loss of important information, so it is usually preferable to combine over-sampling with other techniques. For example, Synthetic Minority Over-sampling Technique (SMOTE) generates synthetic minority class samples based using a nearest neighbors approach [Elreedy and Atiya, 2019]. In practice, these data preprocessing techniques can be used together with our proposed approach and comparator methods. However, the exploration of this is outside the scope of this paper and we leave it to future work.

## 3 COMMON EVENT TETHERING

We consider the use of two models, *common event tethering logistic regression (CET-LR)* and *common event tethering neural network (CET-NN)*, that share information across rare events to improve performance. The sharing of information is encouraged by incorporating a regularization term that penalizes the difference between either the weights in a

logistic regression model or the final layer weights in a neural network.

## 3.1 CET LOGISTIC REGRESSION

In Equation 1, if $h(\mathbf{x}_i) = \mathbf{H}\mathbf{x}_i$, for some matrix $\mathbf{H} \in \mathbb{R}^{d \times p}$, logistic regression is correctly specified to learn the underlying probability models. For such cases, we introduce a common event tethering logistic regression (CET-LR) model.

CET-LR simultaneously maximizes the joint log-likelihood of $\boldsymbol{\theta}_1$ and $\boldsymbol{\theta}_2$ while incorporating a similarity penalty between the parameter vectors. Let $\boldsymbol{\theta} = [\boldsymbol{\theta}_1, \boldsymbol{\theta}_2] \in \mathbb{R}^{2p}$. The log-likelihood of CET-LR is

$$\mathcal{L}^{(s)}(\boldsymbol{\theta}|\mathcal{D}_n) = \mathcal{L}(\boldsymbol{\theta}|\mathcal{D}_n) - \frac{1}{2}s\|\boldsymbol{\theta}_1 - \boldsymbol{\theta}_2\|_2^2. \quad (2)$$

Here, $\mathcal{L}(\boldsymbol{\theta}|\mathcal{D}_n)$ is the unregularized log-likelihood of $\boldsymbol{\theta}$ and $s \geq 0$ is a constant used to control the strength of the similarity regularization term. We note that CET-LR is equivalent to Lapedriza et al. [2007] in the case when $M = 2$.

## 3.2 CET NEURAL NETWORK

We now consider the setting where $h(\mathbf{x}_i)$ maps to a set of latent features that are a non-linear combination of the input features in $\mathbf{x}_i$. As such, a logistic regression model on the input features will be underspecified. To combat this, we introduce common event tethering neural network (CET-NN) as an extension to CET-LR. CET-NN fits an encoder, $\hat{h}_{\phi}$, that maps the input features to a set of latent features. It then uses these latent features as input to a CET-LR model.

The learning of $\hat{h}_{\phi}$ and the $\boldsymbol{\theta}$ parameters of CET-NN are done simultaneously in a standard neural network architecture. In this setup, $\boldsymbol{\theta}$ simply becomes the final layer weight matrix mapping to the outcome vector $\mathbf{y}_i$. In particular, we can write the unregularized log-likelihood as

$$\mathcal{L}(\boldsymbol{\theta}, \boldsymbol{\phi}|\mathcal{D}_n) =$$
$$\sum_{i=1}^{n} \left[ \mathbf{y}_i' \log \left( \begin{bmatrix} \hat{f}_1(\mathbf{x}_i) \\ \hat{f}_2(\mathbf{x}_i) \end{bmatrix} \right) + \right.$$
$$\left. \left( \begin{bmatrix} 1 \\ 1 \end{bmatrix} - \mathbf{y}_i \right)' \log \left( \begin{bmatrix} 1 - \hat{f}_1(\mathbf{x}_i) \\ 1 - \hat{f}_2(\mathbf{x}_i) \end{bmatrix} \right) \right]. \quad (3)$$

where

$$\hat{f}_1(\mathbf{x}_i) = \sigma(\boldsymbol{\theta}_1 \hat{h}_{\phi}(\mathbf{x}_i)), \ \hat{f}_2(\mathbf{x}_i) = \sigma(\boldsymbol{\theta}_2 \hat{h}_{\phi}(\mathbf{x}_i)).$$

Then, the log-likelihood of CET-NN with similarity penalty is

$$\mathcal{L}^{(s)}(\boldsymbol{\theta}, \boldsymbol{\phi}|\mathcal{D}_n) = \mathcal{L}(\boldsymbol{\theta}, \boldsymbol{\phi}|\mathcal{D}_n) - \frac{1}{2}s\|\boldsymbol{\theta}_1 - \boldsymbol{\theta}_2\|_2^2. \quad (4)$$

where $s \geq 0$ is again a constant used to control the strength of the regularization term.

## 3.3 METHOD IMPLEMENTATION

We solve CET-LR and CET-NN by minimizing the negative log-likelihood. Having derived the log-likelihood of CET-LR in Equation 2 and CET-NN in Equation 4, we write the optimization problem in Equation 5.

$$\min_{\boldsymbol{\theta}, \boldsymbol{\phi}} \left[ -\mathcal{L}^{(s)}(\boldsymbol{\theta}, \boldsymbol{\phi}|\mathcal{D}_n) + \text{Reg}(\boldsymbol{\theta}) \right] \quad (5)$$

where $\boldsymbol{\phi}$ is ommitted for CET-LR . $\text{Reg}(\boldsymbol{\theta})$ is used to denote any additional regularization applied to the learned parameters. One may want to employ a general penalty term in addition to the similarity penalty to stabilize performance and further reduce overfitting.

The $L_2$ similarity penalty in Equations 2 and 4 can be replaced with any measure of distance/similarity between two vectors. The choice of this may vary depending on the application. Section 5 presents experimental results using the $L_2$ and $L_1$ magnitude of the difference of $\boldsymbol{\theta}_1$ and $\boldsymbol{\theta}_2$ as well as the cosine-similarity between the two vectors.

Equation 5 can be solved using any applicable optimization approach such as stochastic gradient descent. The parameter $s$ can be learned using validation or set based on pre-existing knowledge of the similarity between the events of interest.

# 4 THEORETICAL RESULTS

In this section, we conduct a theoretical analysis of our proposed method. We first describe a two-step approach to CET-LR and derive asymptotic properties of this estimator. This setup allows us to compare directly to unregularized and ridge LR (Section 4.2). We also use this setup to illustrate the efficiency advantages of utilizing related outcomes that occur more frequently (Section 4.3).

We proceed to derive the asymptotic properties of CET-LR. We establish that this estimator is asymptotically unbiased with a lower asymptotic mean-squared error than LR when the slope parameters of the outcome models are the same and the similarity penalty is greater than zero.

Finally, we discuss how with additional assumptions on the latent features these theoretical results can be extended to CET-NN .

## 4.1 TWO-STEP APPROACH

We consider a two-step variation of LR that incorporates a penalty term between the current parameters being estimated and parameters previously estimated with LR for

another outcome. It is akin to ridge LR where the the estimated parameters are penalized by their distance from the parameters of a similar event, rather than by their distance from zero.

We let $\hat{\boldsymbol{\theta}}_2^{(k)}$ denote our parameter estimates for the more common event that are estimated from ridge LR with penalty parameter $k$. For simplicity, we omit the $k$ superscript from this estimate and let $\hat{\boldsymbol{\theta}}_2 = \hat{\boldsymbol{\theta}}_2^{(k)}$.

We then estimate the parameters $\boldsymbol{\theta}_1$ of our rare event logistic model by maximizing the log-likelihood

$$\mathcal{L}^{(s)}(\boldsymbol{\theta}_1|\mathcal{D}_n, \hat{\boldsymbol{\theta}}_2) = \mathcal{L}(\boldsymbol{\theta}_1|\mathcal{D}_n) - \frac{1}{2}s\|\boldsymbol{\theta}_1 - \hat{\boldsymbol{\theta}}_2\|_2^2, \quad (6)$$

where $\mathcal{L}(\boldsymbol{\theta}_1|\mathcal{D}_n)$ is the unregularized log-likelihood and $s$ is the similarity penalty parameter that controls the strength of the penalization term. We let $\hat{\boldsymbol{\theta}}_1^{(s)}$ denote this estimate but once again omit the $s$ superscript for the majority of this paper, letting $\hat{\boldsymbol{\theta}}_1 = \hat{\boldsymbol{\theta}}_1^{(s)}$.[*]

Like ridge LR, this two-step approach to CET-LR introduces an $L_2$ penalty term into the log-likelihood. In doing so, it maintains the desirable properties of ridge regularization such as avoiding overfitting and handling multicollinearity. However, rather than pulling the coefficients of $\boldsymbol{\theta}_1$ towards zero, it pulls them towards the estimated coefficients of the more common event.

## 4.2 TWO-STEP ASYMPTOTIC PROPERTIES

Theorem 4.1 establishes the asymptotic bias and variance of the two-step CET-LR approach.

**Theorem 4.1** (Two-Step CET-LR Asymptotic Properties). *Let $\mathbb{E}[\hat{\boldsymbol{\theta}}_2]$ and $Var[\hat{\boldsymbol{\theta}}_2]$ be the asymptotic expectation and variance of the estimate of $\boldsymbol{\theta}_2$. Then the MLE estimate of $\mathcal{L}^{(s)}(\boldsymbol{\theta}_1|\mathcal{D}_n, \hat{\boldsymbol{\theta}}_2)$, $\hat{\boldsymbol{\theta}}_1$, has asymptotic bias*

$$\mathbb{E}[\hat{\boldsymbol{\theta}}_1 - \boldsymbol{\theta}_1] = -s\left(\boldsymbol{\Omega}(\boldsymbol{\theta}_1) + s\mathbf{I}\right)^{-1}\left[\boldsymbol{\theta}_1 - \mathbb{E}[\hat{\boldsymbol{\theta}}_2]\right] \quad (7)$$

*and asymptotic variance*

$$Var[\hat{\boldsymbol{\theta}}_1] =$$
$$\left(\boldsymbol{\Omega}(\boldsymbol{\theta}_1) + s\mathbf{I}\right)^{-1}\left(\boldsymbol{\Omega}(\boldsymbol{\theta}_1) + s^2 Var[\hat{\boldsymbol{\theta}}_2]\right)\left(\boldsymbol{\Omega}(\boldsymbol{\theta}_1) + s\mathbf{I}\right)^{-1}. \quad (8)$$

*Here, $\boldsymbol{\Omega}(\cdot)$ is the negative of the hessian matrix and $\boldsymbol{\theta}_1$ is the true parameter vector of event 1. $\mathbf{I}$ is a $p \times p$ identity matrix.*

We observe from Equation 7 in Theorem 4.1 that, compared to ridge LR, two-step CET-LR can decrease the asymptotic bias of the estimated parameter vector of event 1 ($\hat{\boldsymbol{\theta}}_1$) if

it's true parameter vector ($\boldsymbol{\theta}_1$) is closer to the asymptotic expectation of event 2 ($\mathbb{E}[\hat{\boldsymbol{\theta}}_2]$) than it is to the zero vector.[*] In particular, we note that the estimate of two-step CET-LR is asymptotically unbiased if the true parameters of event 1 ($\boldsymbol{\theta}_1$) equal the asymptotic expectation of the estimated parameters of event 2 ($\mathbb{E}[\hat{\boldsymbol{\theta}}_2]$).

However, two-step CET-LR does incur slightly higher variance from using the estimates of $\boldsymbol{\theta}_2$ in its regularization term.[*] We quantify when this exchange of decreased bias for higher variance is beneficial by comparing the asymptotic mean-squared error (MSE) of the two-step CET-LR estimator versus ridge LR in Theorem 4.2.

**Theorem 4.2** (Two-Step CET-LR vs. Ridge LR MSE). *Let $\tilde{\boldsymbol{\theta}}_1$ be the ridge LR estimate of $\boldsymbol{\theta}_1$ with ridge penalty parameter $s$. And let $\hat{\boldsymbol{\theta}}_1$ be the two-step CET-LR estimate with similarity parameter also $s$ and $\hat{\boldsymbol{\theta}}_2$ the estimate for event 2 used in the penalty term. As in Theorem 4.1, let $\mathbb{E}[\hat{\boldsymbol{\theta}}_2]$ be the asymptotic expectation of $\hat{\boldsymbol{\theta}}_2$.*

*We let $\boldsymbol{\theta}_1$ and $\boldsymbol{\theta}_2$ be the true parameter vectors for events 1 and 2 respectively, and assume that there exists an orthogonal matrix $\mathbf{P}$ such that $\boldsymbol{\Omega}(\boldsymbol{\theta}_1) = \mathbf{P}\mathbf{A}\mathbf{P}'$ and $\boldsymbol{\Omega}(\boldsymbol{\theta}_2) = \mathbf{P}\mathbf{B}\mathbf{P}'$ for diagonal matrices $\mathbf{A}$ and $\mathbf{B}$.*

*We then let $\mathbf{a} = \mathbf{P}\boldsymbol{\theta}_1$ and $\mathbf{b} = \mathbf{P}\mathbb{E}[\hat{\boldsymbol{\theta}}_2]$ be the projections of $\boldsymbol{\theta}_1$ and $\mathbb{E}[\hat{\boldsymbol{\theta}}_2]$ onto the column space of $\mathbf{P}$.*

*Denoting MSE as the asymptotic mean-squared error of an estimator, we find that*

$$MSE\left(\hat{\boldsymbol{\theta}}_1\right) < MSE\left(\tilde{\boldsymbol{\theta}}_1\right) \quad (9)$$

*when*

$$b_j\left(2a_j - b_j\right) > \frac{B_{j,j}}{(B_{j,j} + k)^2} \quad (10)$$

*for all $j \in \{1, p\}$.*

*The above is a sufficient, but not necessary, condition. If we denote the left-hand side of Equation 10 as $\eta_j$ and the right-hand side as $\beta_j$, and further let $\alpha_j = \frac{1}{(A_{j,j}+s)^2}$, a more relaxed condition sufficient to imply Equation 9 is that*

$$\sum_{j=1}^{p} \alpha_j \eta_j > \sum_{j=1}^{p} \alpha_j \beta_j. \quad (11)$$

Theorem 4.2 establishes a relationship between the reduced bias and added variance of using estimated parameter values of a related event as a baseline for regularization. In essence, this theorem shows that the degree to which the parameter vector for event 1 is closer to the parameter vector for event

---

[*]We mention this superscript notation for use in Section 4.3.

[*]The asymptotic bias of $\hat{\boldsymbol{\theta}}_1$ estimated with ridge LR and penalty parameter $s$ is $\mathbb{E}[\hat{\boldsymbol{\theta}}_1 - \boldsymbol{\theta}_1] = -s\left(\boldsymbol{\Omega}(\boldsymbol{\theta}_1) + s\mathbf{I}\right)^{-1}\boldsymbol{\theta}_1$.

[*]The asymptotic variance of $\hat{\boldsymbol{\theta}}_1$ estimated with ridge LR and penalty parameter $s$ is $Var[\hat{\boldsymbol{\theta}}_1] = \left(\boldsymbol{\Omega}(\boldsymbol{\theta}_1) + s\mathbf{I}\right)^{-1}\boldsymbol{\Omega}(\boldsymbol{\theta}_1)\left(\boldsymbol{\Omega}(\boldsymbol{\theta}_1) + s\mathbf{I}\right)^{-1}$.

2 than it is to the zero vector must be enough to outweigh the added variance of estimating the parameters for event 2. In practice, this suggests that a tethering approach can be beneficial when a more common event's parameters can be estimated with low variance and are believed to be similar to the parameters for a rare event of interest. However, tethering to a more common event may not be a good idea if the variance of the common event's estimated parameters is large. We include the proof for Theorem 4.2 and expand further on its implications in Appendix A

The formalization of just how close $\boldsymbol{\theta}_1$ needs to be to $\mathbb{E}[\hat{\boldsymbol{\theta}}_2]$ is made difficult due to the complex behavior of MLE solvers. Using intuition drawn from [Keskar et al., 2016], one can view the eigenvalues (i.e. diagonal values of $\mathbf{A}$ and $\mathbf{B}$) as characterizing the sharpness of the minimizer along its corresponding eigenvector in $\mathbf{P}$. In this case, for CET-LR to improve upon ridge RL, $\mathbf{a}$ needs to be closer to $\mathbf{b}$ than $\mathbf{0}$, particularly in the crucial directions for loss minimization.

### 4.3 FINITE SAMPLE EFFICIENCY

It has long been understood that rare event prediction is hindered by the lack of positive samples. Such behavior can be understood by observing the asymptotic efficiency of MLE estimators for rare events. Whereas the unregularized MLE of LR converges at a rate of $n^{-\frac{1}{2}}$, Wang [2020] showed that the the convergence rate of $\hat{\boldsymbol{\theta}}_1$ is $O_p(n_1^{-\frac{1}{2}})$, where $n_1 = \sum_{i=1}^n y_{i,1}$. To better understand the benefit of sharing information between rare events in finite samples we decompose the two-step CET-LR into two components. To do so, we reintroduce the superscript $s$ into the estimate $\hat{\boldsymbol{\theta}}_1^{(s)}$ and let $\hat{\boldsymbol{\theta}}_1$ and $\hat{\boldsymbol{\theta}}_2$ denote the unregularized LR estimates of the parameter vectors. With this notation, we can write

$$\hat{\boldsymbol{\theta}}_1^{(s)} = \left(\boldsymbol{\Omega}^{(s)}(\boldsymbol{\theta}_1)\right)^{-1} \boldsymbol{\Omega}(\boldsymbol{\theta}_1)\hat{\boldsymbol{\theta}}_1 + s \left(\boldsymbol{\Omega}^{(s)}(\boldsymbol{\theta}_1)\right)^{-1} \hat{\boldsymbol{\theta}}_2 \tag{12}$$

where $\boldsymbol{\Omega}^{(s)}(\boldsymbol{\theta}_1) = \boldsymbol{\Omega}(\boldsymbol{\theta}_1) + s\mathbf{I}$.

If we let $n_2 = \sum_{i=1}^n y_{i,2}$ and assume that $n_2 > n_1$, then there is more information in $\mathcal{D}_n$ about the more common outcome, $y_{i,2}$, than the rarer outcome, $y_{i,1}$. In terms of convergence rates, we note that $\hat{\boldsymbol{\theta}}_2 = O_p(n_2^{-\frac{1}{2}})$ converges faster than $\hat{\boldsymbol{\theta}}_1 = O_p(n_1^{-\frac{1}{2}})$. We further observe that as $s$ increases, the coefficient values of $\hat{\boldsymbol{\theta}}_1^{(s)}$ becomes more strongly composed of $\hat{\boldsymbol{\theta}}_2$ than $\hat{\boldsymbol{\theta}}_1$. From here, we see how, when we believe $\boldsymbol{\theta}_1$ and $\boldsymbol{\theta}_2$ to be similar, imposing a strong similarity regularization term can help with faster convergence and ultimately result in better performance on real-world datasets.

### 4.4 CET-LR ASYMPTOTIC PROPERTIES

We derive the asymptotic properties of the CET-LR estimator in Appendix A.1 which we use to prove Theorem 4.3.

**Theorem 4.3** (CET-LR Asymptotic MSE). *If $\boldsymbol{\theta}_1 = \boldsymbol{\theta}_2$, then for any $s' > s \geq 0$ the asymptotic MSE of the MLE estimate of $\boldsymbol{\theta} = [\boldsymbol{\theta}_1, \boldsymbol{\theta}_2]$ is less under the log-likelihood of $\mathcal{L}^{(s')}(\boldsymbol{\theta}|\mathcal{D}_n)$ than $\mathcal{L}^{(s)}(\boldsymbol{\theta}|\mathcal{D}_n)$.*

Theorem 4.3 shows that when $\boldsymbol{\theta}_1 = \boldsymbol{\theta}_2$ CET-LR is a better estimator than unregularized LR (i.e. $s = 0$) and continues to improve as $s$ grows. In fact, as $s \to \infty$, the asymptotic MSE of CET-LR for $\boldsymbol{\theta} = [\boldsymbol{\theta}_1, \boldsymbol{\theta}_2]$ approaches that of LR for just $\boldsymbol{\theta}_1$.[*]

### 4.5 EXTENSION TO CET-NN

Theoretical analysis of feature learning in neural networks is notoriously challenging and beyond the scope of this work. However, we note that the preceding theoretical analyses apply not only in the linear case, but also in the case where nonlinear DGP features are faithfully recovered (up to permutation), as this assumption reduces learning to logistic regression on latent rather than raw features. Thus, understanding differences in benefits of CET-LR and CET-NN depends on a theoretical account of feature learning in the MLL setting.

## 5 SIMULATIONS

In this section, we conduct a simulation study to test the performance of CET-LR and CET-NN under a wide range of settings. Our goal is to determine how the performance of CET-LR and CET-NN are affected by (i) the underlying data generation process (DGP), (ii) the degree of similarity between the rare and common outcomes, and (iii) the event rate of the common outcome. We outline the linear and non-linear DGPs we use in these simulations and other simulation setups details in Section 5.1. We then explore the impact of varying event similarity in Section 5.2 and varying common outcome event rate in Section 5.3. Our results provide a foundation for understanding the conditions under which CET methods provide additional benefit and set the stage for the use of these methods in real-world clinical applications (Section 6).[*]

### 5.1 SETUP

Both our linear and non-linear DGPs are designed to simulate a setup where 25 input patient features are used to

---

[*]See the proof of Theorem 4.3 in Appendix A for the calculation of the asymptotic MSE of CET-LR .

[*]Code to run experiments available at https://github.com/engelhard-lab/rare_event_mll.

predict a rare disease of interest, $y_{i,1}$, whose underlying risk function is related to a more common disease, $y_{i,2}$. The degree of similarity between the underlying risk functions for $y_{i,1}$ and $y_{i,2}$ is a controllable parameter passed as an argument to the DGP. For details on both the linear and non-linear DGP setups see Appendix B.1.

For all experiments using the linear DGP, we generate a training set of 15,000 synthetic patients. We increase the number of training samples to 75,000 for the non-linear DGP. In addition to the $L_2$ similarity penalty in Equations 2 and 4, we consider a corresponding $L_1$ variety as well as a version that uses cosine similarity.

We compare CET-LR and CET-NN to single-label learning trained exclusively on $y_{i,1}$, and standard MLL without similarity penalty (*i.e.*, CET method with $s = 0$). All methods use standard ridge (*i.e.*, $L_2$) regularization to avoid overfitting. Both $s$ and the regularization strength parameter are optimized by grid search on a validation set. See Appendix B.2 for further training details.

We use a large test set of 35,000 synthetic patients to reduce the uncertainty in the evaluation stemming from the rarity of the outcomes. We adopt the evaluation method described in Kmetzsch et al. [2022], which uses Spearman's rank correlation ($\rho$) as the primary metric on simulated data to assess the discrepancy between predicted and actual disease risk rankings. We include results using AUC as the evaluation metric in Appendix C.2, as it serves as a proxy for $\rho$ in real-world settings where true risk is unknown.

In Appendix C.1, we include additional results comparing CET approaches to Firth regression, gradient boosting, and transfer learning. These methods provide alternative baselines for rare event prediction and are discussed further in the Appendix.

## 5.2 VARYING EVENT SIMILARITY

We first explore how the benefit of CET is affected by event similarity by varying the risk function similarity between the rare and common event from 0% to 100%. We set the expected event rate to 1% for the rare disease and to 5% for the more common disease. Simulations are run using both the linear and non-linear DGPs.

Figure 1a shows how CET-LR enhances rare disease prediction under the linear DGP at different similarity levels. All three variants of CET-LR outperform the baseline single-label learning when event similarity exceeds a certain threshold (∼40%), and this improvement grows as event similarity increases. Note that in contrast to the non-linear setting (Figure 1b), the standard MLL without a similarity penalty has no benefit, because no information is shared between labels without a similarity regularization term.

Figure 1b shows the results for the same experiment using

the non-linear DGP, where we see similar results for CET-NN as we did for CET-LR on the linear DGP. However, in this setup, the standard MLL approach (i.e. no similarity penalty) does improve upon single-label learning, a result that is consistent with the literature on MLL with neural networks. However, we note that the similarity penalty can result in further improvement, specifically when there is sufficient overlap of the latent features and their corresponding weights (*e.g.*, $\geq 40\%$).

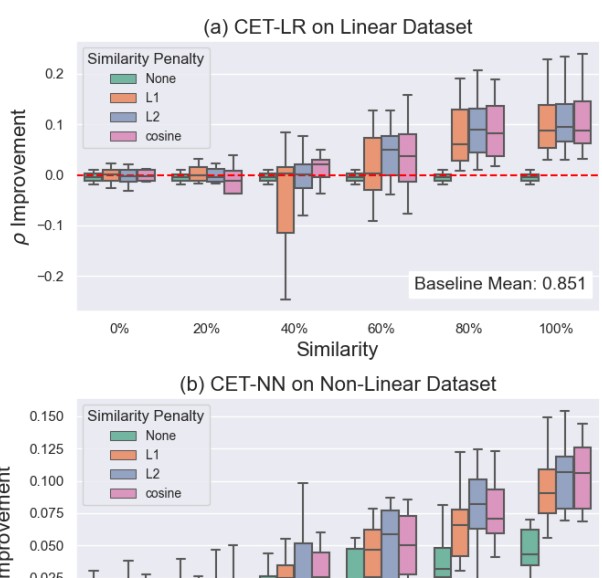

Figure 1: Boxplots representing pairwise enhancement for rare disease prediction based on 10 iterations for each event similarity setting. The red line indicates the baseline of single-label learning for each iteration.

We further analyze the similarity penalty parameter ($s$) selected via validation versus the underlying event similarity in Appendix C.3. The positive correlation supports the claim that the improved performance of the CET methods is coming from the added similarity penalty.

## 5.3 VARYING EVENT RATE

We now explore how the benefit of CET is affected by the event rate of the more common event. To do so, we hold the event similarity and rare outcome event rate constant at 80% and 1%, respectively. We then vary the common outcome event rate from 1% to 30%.

Figure 2a shows that increasing the event rate of $y_{i,2}$ not only improves predictive performance for $y_{i,2}$, but also provides substantial improvement for $y_{i,1}$. We observe a similar trend

in non-linear settings (Appendix C.4), supporting the claim that CET methods can leverage the additional information in a dataset for a more common event to help overcome the lack of information for a rarer event.

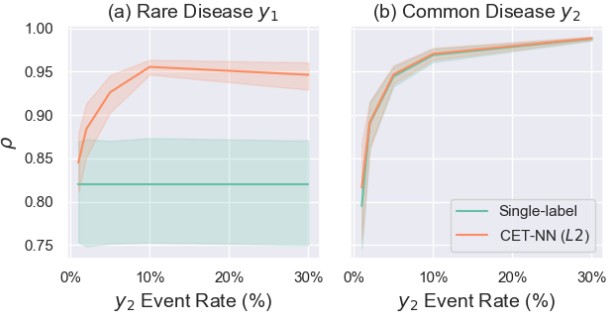

Figure 2: Performance of single-label learning and CET-LR on (a) rare and (b) common diseases generated by the linear DGP. The rare disease ($y_1$) event rate is 1%, and the common disease ($y_2$) event rate is varied from 1% to 30%. Shading represents 95% confidence intervals.

# 6    REAL-WORLD EXPERIMENTS

The simulation results in Section 5 demonstrate the effectiveness of CET in settings with sufficient event similarity and common event rate. In this section, we implement our CET approach on two real-world datasets, analyzing the extent to which their ability to leverage information from a more common outcome can facilitate better performance.

Our two datasets are comprised of electronic health record (EHR) data. The first comes from a preeclampsia study on women with hypertensive disorders of pregnancy [Meng et al., 2023] and the second comes from an early autism study on children under the age of 18 months. We are interested in using each dataset to train a prognostic model for a rare outcome and wish to leverage similar alternative patient outcomes.

Similar to Section 5, we compare CET-LR and CET-NN to a single-label learning baseline and MLL without a similarity penalty. For both datasets, we use a validation set to perform early stopping and hyperparameter tuning. We employ AUC as our performance metric and show results on the rare outcome of interest. This section shows results for NN models due to their superior performance. See Appendix B.2 for more implementation details and Appendix C.5 for results using LR models.

## 6.1    MATERNAL MORBIDITY IN PREECLAMPSIA

The maternal morbidity dataset is composed of 553,658 patients with hypertensive disorders of pregnancy. The input features include patient demographics and ICD code-based diagnoses and medical procedures as well as hospital-level characteristics. The dataset contains four binary outcomes denoting whether a rare morbidity event occurred within one-year post-delivery. These outcomes are stroke (event rate 0.075%), hypertensive crisis (0.193%), heart failure (0.248%), and acute renal failure (0.171%). Stroke has lowest event rate and high clinical importance, therefore we select it as our primary outcome of interest. We separately assess the benefit of using each of the remaining outcomes as our common event, and discuss the possibility of adopting an architecture that uses all four outcomes together in Section 7.

This dataset is unusually large compared to single-institution clinical datasets more commonly used to train risk prediction models due to the complexities of sharing medical data across sites. To align more closely with such datasets, we randomly sampled a subset of 80,000 patients for model training and allocated the remaining patients to be used for evaluation. In doing so, we are also able to mitigate the unstable performance metrics that can often be produced with small test sets for rare events.

Figure 3 shows that CET-NN consistently outperforms the baseline when each of the three other morbidity outcomes are used as the common event, with the most significant improvement observed when using hypertensive crisis. Given the similar event rates among the three common outcomes we considered, the increased benefit of using hypertensive crisis suggests a high degree of similarity between its risk factors and the risk factors of stroke. This aligns with clinical understanding of the strong link between hypertensive crisis and stroke [Pistoia et al., 2016].

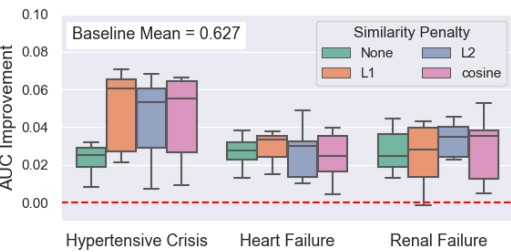

Figure 3: Boxplots showing pairwise improvement in the AUC of stroke prediction via CET-NN across 10 iterations. The red line indicates the single-label learning baseline.

## 6.2    EARLY AUTISM PREDICTION

The early autism dataset contains medical information on 18,156 children. Features for a given patient were derived by first extracting each diagnosis and procedure code documented in that patient's chart from birth to 18 months, then mapping them to corresponding 256-dimensional word2vec embeddings. The resulting diagnosis and procedure embeddings were mean-pooled, and the resulting vectors were

concatenated to a single 512-dimensional feature vector. The outcomes are comprised of multiple neurodevelopmental diagnoses (ND) including ADHD, developmental delay, language delay, motor delay, and autism. We select autism as the target outcome (event rate 2.2%) and define the presence of any other ND as the common event (18.5%). We also explore the effect of splitting common event to single ND in the Appendix C.7. We divided the data into training and testing sets with a ratio of 4:1, and implemented the same model training and validation strategies as in previous experiments.

The result shows that incorporating more common ND outcomes into MLL models via CET-NN (Figure 4) or CET-LR (Appendix C.5) significantly enhances autism prediction performance. This indicates a substantial similarity in features or latent features across various ND outcomes.

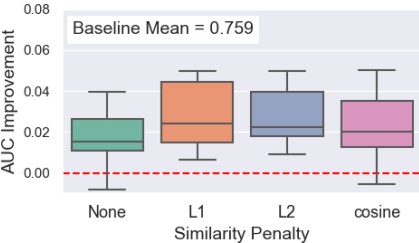

Figure 4: Boxplots showing pairwise improvement in the AUC of autism prediction via CET-NN across 10 iterations. The red line indicates the single-label learning baseline.

## 7  CONCLUSION

We propose the use of common event tethering as a variation of regularized MLL optimized for rare event modeling. Our proposed methods, CET-LR and CET-NN, build on existing literature by coupling the learning of shared features via neural network with a regularization approach that shrinks rare event coefficients toward those of a more common event using several alternative measures of vector similarity. We provide rigorous supporting theoretical and empirical analyses showing the conditions under which CET methods are beneficial and exploring how leveraging more common events can lead to faster convergence rates. We support our findings with results on two real-world medical applications; first predicting rare cardiovascular morbidities in pregnant people with HDP and then predicting autism likelihood in early childhood. We provide proofs to our theoretical results as well as additional experimental results in the Appendix.

We conclude this paper with a brief commentary of important considerations when implementing our CET approach. We provide insight into *finding surrogate events*, comment on important *ethical considerations*, address the *case when M > 2*, and outline *limitations and future work*.

**Finding Surrogate Events**    In our real-world experiments, we saw greatest improvement in predicting a rare event (stroke) when we used a more common event (hypertensive crisis) that is known to be physiologically related and shares clinical risk factors [Pistoia et al., 2016]. In contrast, our other real-world examples used rare and common event combinations without a similarly strong known physiological link. This finding suggests that previous literature and domain knowledge should inform selection of suitable surrogate events.

**Ethical Considerations and Bias**    In this paper, we show the potential performance gains of tethering rare events to related, more common events. However, we note that it is important for researchers to consider biases in a given clinical context and setting that may be relevant to the use of CET. The naive use of CET or other MLL approaches could worsen existing biases by propagating bias from one (biased) outcome to another (less biased) outcome.

For example, both autism and ADHD are more common in boys than girls, but the imbalance is greater for autism, and girls tend to be diagnosed less often and at a later age [Loomes et al., 2017]. Therefore, tethering ADHD to autism could lead to a more biased model compared to training using ADHD outcomes alone.

**More Than Two Outcomes**    We explored using all four outcomes to help predict stroke in the maternal morbidity dataset by including a penalty for all pairs of coefficients. In this setup, we saw negligible improvement, as the AUC went from 0.670 without the CET penalty to 0.674 with the CET penalty (full results included in Appendix C.5). We hypothesize that including additional outcomes is more forgiving for feature learning improvement but is less beneficial for similarity-based penalty approaches when all the outcomes are not closely related, as is the case in the maternal morbidity dataset (see the pairwise comparisons in Figure 3).

The baseline approach for $M > 2$ incorporates a penalty for each pair of outcome coefficient vectors. However, with domain knowledge the framework could be modified to only incorporate penalty terms between the rare event and a select number of the most similar common events. We leave a more detailed exploration of this topic to future work. At this time, we advocate for targeted selection of a limited number of outcomes with related clinical etiology, especially when you consider the previously discussed risks of common event tethering .

**Limitations & Future Work**    The primary limitation of CET-LR and CET-NN is that using it effectively requires domain knowledge (*i.e.*, clinical expertise) to select common events that share risk factors with rare events of interest. Results show that under typical conditions, our approach

does not worsen performance even when the common and rare events are unrelated. Nevertheless, we believe standard MLL approaches may be more appropriate when such domain knowledge is not available.

Additionally, CET-LR and CET-NN effectiveness depends on a reasonable choice of the similarity penalty parameter $s$. While validation or domain knowledge can guide the choice of $s$, poorly chosen $s$ values may lead to suboptimal results.

Finally, whereas the current work does explore empirically the interaction betweeen feature learning and our proposed regularization term, in future work we will more rigorously explore whether benefits of CET-NN depend on the number of latent features (*i.e.*, hidden layer width). We hypothesize that CET-NN provides greater benefit in the neural tangent kernel regime (*i.e.*, wide hidden layer) [Jacot et al., 2018] and less benefit in the feature learning regime (*i.e.*, narrow hidden layer).

## ACKNOWLEDGEMENTS

Funding support was provided by the Foundation for Anesthesia Education and Research (FAER). ME is supported by grant K01-MH127309 from the National Institute of Mental Health (NIMH). QL is supported by National Science Foundation (NSF) grant DMS-2046880. QL also thanks the NSF Artificial Intelligence for Designing and Understanding Materials - National Research Traineeship (aiM-NRT) at Duke University funded under grant DGE-2022040.

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

# Common Event Tethering to Improve Prediction of Rare Clinical Events (Supplementary Material)

Quinn Lanners*[1]    Qin Weng*[1]    Marie-Louise Meng[2]    Matthew M. Engelhard[1]

[1]Department of Biostatistics & Bioinformatics, Duke University School of Medicine, Durham, NC, USA
[2]Department of Anesthesiology, Duke University School of Medicine, Durham, NC, USA

## A  PROOFS FOR THEOREMS IN SECTION 4

**Theorem 4.1** (Two-Step CET-LR Asymptotic Properties). *Let $\mathbb{E}[\hat{\boldsymbol{\theta}}_2]$ and Var$[\hat{\boldsymbol{\theta}}_2]$ be the asymptotic expectation and variance of the estimate of $\boldsymbol{\theta}_2$. Then the MLE estimate of $\mathcal{L}^{(s)}(\boldsymbol{\theta}_1|\mathcal{D}_n, \hat{\boldsymbol{\theta}}_2)$, $\hat{\boldsymbol{\theta}}_1$, has asymptotic bias*

$$\mathbb{E}[\hat{\boldsymbol{\theta}}_1 - \boldsymbol{\theta}_1] = -s\left(\boldsymbol{\Omega}(\boldsymbol{\theta}_1) + s\mathbf{I}\right)^{-1}\left[\boldsymbol{\theta}_1 - \mathbb{E}[\hat{\boldsymbol{\theta}}_2]\right] \tag{7}$$

*and asymptotic variance*

$$Var[\hat{\boldsymbol{\theta}}_1] =$$
$$(\boldsymbol{\Omega}(\boldsymbol{\theta}_1) + s\mathbf{I})^{-1}\left(\boldsymbol{\Omega}(\boldsymbol{\theta}_1) + s^2 Var[\hat{\boldsymbol{\theta}}_2]\right)(\boldsymbol{\Omega}(\boldsymbol{\theta}_1) + s\mathbf{I})^{-1}. \tag{8}$$

*Here, $\boldsymbol{\Omega}(\cdot)$ is the negative of the hessian matrix and $\boldsymbol{\theta}_1$ is the true parameter vector of event 1. $\mathbf{I}$ is a $p \times p$ identity matrix.*

*Proof.* We use an approach and notation similar to that of Cessie and Houwelingen [1992]. In particular, we use the Newton-Raphson maximization procedure to arrive at these asymptotic properties. Let $\mathcal{L}^{(s)}(\boldsymbol{\theta}_1) = \mathcal{L}^{(s)}(\boldsymbol{\theta}_1|\mathcal{D}_n, \hat{\boldsymbol{\theta}}_2)$ for simplicity. Before proceeding, we add the superscript $k$ to the ridge estimate of $\boldsymbol{\theta}_2$ with ridge penalty parameter $k$, making it $\hat{\boldsymbol{\theta}}_2^{(k)}$. This is done so we can allow $\hat{\boldsymbol{\theta}}_2$ to denote the unregularized estimate. We do a similar thing for the estimate of $\boldsymbol{\theta}_1$ from CET-LR , letting $\hat{\boldsymbol{\theta}}_1^{(s)}$ be the regularized estimate and $\hat{\boldsymbol{\theta}}_1$ be the unregularized estimate.

We take the first derivative of $\mathcal{L}^{(s)}(\boldsymbol{\theta}_1)$

$$\mathbf{U}^{(s)}(\boldsymbol{\theta}_1) = \mathbf{U}(\boldsymbol{\theta}_1) - s(\boldsymbol{\theta}_1 - \hat{\boldsymbol{\theta}}_2^{(k)}).$$

$\mathbf{U}(\boldsymbol{\theta}_1)$ is the first derivative of the unregularized $\mathcal{L}(\boldsymbol{\theta}_1)$.

We then compute the negative hessian matrix,

$$\boldsymbol{\Omega}^{(s)}(\boldsymbol{\theta}_1) = \boldsymbol{\Omega}(\boldsymbol{\theta}_1) + s\mathbf{I}$$

.

We now derive large sample properties of our estimate, $\hat{\boldsymbol{\theta}}_1^{(s)}$, using Taylor series expansion about the true parameter, $\boldsymbol{\theta}_1$. We have

$$\mathbf{U}^{(s)}(\hat{\boldsymbol{\theta}}_1^{(s)}) = \mathbf{U}^{(s)}(\boldsymbol{\theta}_1) - (\hat{\boldsymbol{\theta}}_1^{(s)} - \boldsymbol{\theta}_1)\boldsymbol{\Omega}^{(s)}(\boldsymbol{\theta}_1) + o\left(\|\hat{\boldsymbol{\theta}}_1^{(s)} - \boldsymbol{\theta}_1\|\right).$$

We then arrive at a first-order approximation of $\hat{\boldsymbol{\theta}}_1^{(s)}$

$$\hat{\boldsymbol{\theta}}_1^{(s)} = \boldsymbol{\theta}_1 + (\boldsymbol{\Omega}(\boldsymbol{\theta}_1) + s\mathbf{I})^{-1} \left( \mathbf{U}(\boldsymbol{\theta}_1) - s(\boldsymbol{\theta}_1 - \hat{\boldsymbol{\theta}}_2^{(k)}) \right).$$

Now, as stated in Cessie and Houwelingen [1992], we have that the first order estimate $\boldsymbol{\theta}_1$ that maximizes the unregularized log-likelihood is $\hat{\boldsymbol{\theta}}_1 = \boldsymbol{\theta}_1 + \boldsymbol{\Omega}^{-1}(\boldsymbol{\theta}_1)\mathbf{U}(\boldsymbol{\theta}_1)$. And under certain regularity conditions, we have that $\hat{\boldsymbol{\theta}}_1$ is asymptotically unbiased with covariance matrix $\boldsymbol{\Omega}(\boldsymbol{\theta}_1)^{-1}$. Cessie and Houwelingen [1992] also show that the ridge LR estimate of the true $\boldsymbol{\theta}_2$ with ridge penalty parameter $k$ is

$$\hat{\boldsymbol{\theta}}_2^{(k)} = (\boldsymbol{\Omega}(\boldsymbol{\theta}_2) + k\mathbf{I})^{-1} \boldsymbol{\Omega}(\boldsymbol{\theta}_2)\hat{\boldsymbol{\theta}}_2$$

where $\hat{\boldsymbol{\theta}}_2$ is again the estimate from unregularized logistic regression. From here, we get that the asymptotic bias of $\hat{\boldsymbol{\theta}}_2^{(k)}$ is

$$-k\left(\boldsymbol{\Omega}(\boldsymbol{\theta}_2) + k\mathbf{I}\right)^{-1} \boldsymbol{\theta}_2$$

and asymptotic variance is

$$\left(\boldsymbol{\Omega}(\boldsymbol{\theta}_2) + k\mathbf{I}\right)^{-1} \boldsymbol{\Omega}(\boldsymbol{\theta}_2) \left(\boldsymbol{\Omega}(\boldsymbol{\theta}_2) + k\mathbf{I}\right)^{-1}.$$

Using these properties, we perform the following calculations to arrive at our asymptotic bias.

$$
\begin{aligned}
\mathbb{E}[\hat{\boldsymbol{\theta}}_1^{(s)} - \boldsymbol{\theta}_1] &= \mathbb{E}\left[ \boldsymbol{\theta}_1 + (\boldsymbol{\Omega}(\boldsymbol{\theta}_1) + s\mathbf{I})^{-1} \left( \mathbf{U}(\boldsymbol{\theta}_1) - s(\boldsymbol{\theta}_1 - \hat{\boldsymbol{\theta}}_2^{(k)}) \right) - \boldsymbol{\theta}_1 \right] \\
&= \mathbb{E}\left[ (\boldsymbol{\Omega}(\boldsymbol{\theta}_1) + s\mathbf{I})^{-1} \left( \mathbf{U}(\boldsymbol{\theta}_1) - s(\boldsymbol{\theta}_1 - \hat{\boldsymbol{\theta}}_2^{(k)}) \right) \right] \\
&= -s\left(\boldsymbol{\Omega}(\boldsymbol{\theta}_1) + s\mathbf{I}\right)^{-1} \left[ \boldsymbol{\theta}_1 - \mathbb{E}[\hat{\boldsymbol{\theta}}_2^{(k)}] \right].
\end{aligned}
$$

To derive the variance, we first rewrite our estimate as

$$\hat{\boldsymbol{\theta}}_1^{(s)} = (\boldsymbol{\Omega}(\boldsymbol{\theta}_1) + s\mathbf{I})^{-1} \left( \boldsymbol{\Omega}(\boldsymbol{\theta}_1)\hat{\boldsymbol{\theta}}_1 + s\hat{\boldsymbol{\theta}}_2^{(k)} \right).$$

From here, we can derive the asymptotic variance as shown below.

$$
\begin{aligned}
\mathrm{Var}(\hat{\boldsymbol{\theta}}_1^{(s)}) &= \mathrm{Var}\left[ (\boldsymbol{\Omega}(\boldsymbol{\theta}_1) + s\mathbf{I})^{-1} \left( \boldsymbol{\Omega}(\boldsymbol{\theta}_1)\hat{\boldsymbol{\theta}}_1 + s\hat{\boldsymbol{\theta}}_2^{(k)} \right) \right] \\
&= \mathrm{Var}\left[ (\boldsymbol{\Omega}(\boldsymbol{\theta}_1) + s\mathbf{I})^{-1} \boldsymbol{\Omega}(\boldsymbol{\theta}_1)\hat{\boldsymbol{\theta}}_1 + s\left(\boldsymbol{\Omega}(\boldsymbol{\theta}_1) + s\mathbf{I}\right)^{-1} \left(\boldsymbol{\Omega}(\boldsymbol{\theta}_2) + k\mathbf{I}\right)^{-1} \boldsymbol{\Omega}(\boldsymbol{\theta}_2)\hat{\boldsymbol{\theta}}_2 \right]
\end{aligned}
$$

Now, let

$$A = (\boldsymbol{\Omega}(\boldsymbol{\theta}_1) + s\mathbf{I})^{-1} \boldsymbol{\Omega}(\boldsymbol{\theta}_1)\hat{\boldsymbol{\theta}}_1$$

and let

$$B = s\left(\boldsymbol{\Omega}(\boldsymbol{\theta}_1) + s\mathbf{I}\right)^{-1} \left(\boldsymbol{\Omega}(\boldsymbol{\theta}_2) + k\mathbf{I}\right)^{-1} \boldsymbol{\Omega}(\boldsymbol{\theta}_2)\hat{\boldsymbol{\theta}}_2.$$

Then, we can calculate $\mathrm{Var}(A + B)$. Note that the unregularized estimate $\hat{\boldsymbol{\theta}}_1$ is independent of the unregularized estimate $\hat{\boldsymbol{\theta}}_2$. Therefore, $\mathrm{Var}(A + B) = (\mathrm{Var}(A) + \mathrm{Var}(B))$. We conclude

$$
\begin{aligned}
\mathrm{Var}(\hat{\boldsymbol{\theta}}_1^{(s)}) &= (\boldsymbol{\Omega}(\boldsymbol{\theta}_1) + s\mathbf{I})^{-1} \boldsymbol{\Omega}(\boldsymbol{\theta}_1) (\boldsymbol{\Omega}(\boldsymbol{\theta}_1) + s\mathbf{I})^{-1} + \\
&\quad s^2 \left(\boldsymbol{\Omega}(\boldsymbol{\theta}_1) + s\mathbf{I}\right)^{-1} \left(\boldsymbol{\Omega}(\boldsymbol{\theta}_2) + k\mathbf{I}\right)^{-1} \boldsymbol{\Omega}(\boldsymbol{\theta}_2) \left(\boldsymbol{\Omega}(\boldsymbol{\theta}_2) + k\mathbf{I}\right)^{-1} \left(\boldsymbol{\Omega}(\boldsymbol{\theta}_1) + s\mathbf{I}\right)^{-1} \\
&= (\boldsymbol{\Omega}(\boldsymbol{\theta}_1) + s\mathbf{I})^{-1} \boldsymbol{\Omega}(\boldsymbol{\theta}_1) (\boldsymbol{\Omega}(\boldsymbol{\theta}_1) + s\mathbf{I})^{-1} + s^2 \left(\boldsymbol{\Omega}(\boldsymbol{\theta}_1) + s\mathbf{I}\right)^{-1} \mathrm{Var}[\hat{\boldsymbol{\theta}}_2^{(k)}] \left(\boldsymbol{\Omega}(\boldsymbol{\theta}_1) + s\mathbf{I}\right)^{-1} \\
&= (\boldsymbol{\Omega}(\boldsymbol{\theta}_1) + s\mathbf{I})^{-1} \left( \boldsymbol{\Omega}(\boldsymbol{\theta}_1) + s^2 \mathrm{Var}[\hat{\boldsymbol{\theta}}_2^{(k)}] \right) \left(\boldsymbol{\Omega}(\boldsymbol{\theta}_1) + s\mathbf{I}\right)^{-1}.
\end{aligned}
$$

$\square$

**Theorem 4.2** (Two-Step CET-LR vs. Ridge LR MSE). *Let $\tilde{\boldsymbol{\theta}}_1$ be the ridge LR estimate of $\boldsymbol{\theta}_1$ with ridge penalty parameter $s$. And let $\hat{\boldsymbol{\theta}}_1$ be the two-step CET-LR estimate with similarity parameter also $s$ and $\hat{\boldsymbol{\theta}}_2$ the estimate for event 2 used in the penalty term. As in Theorem 4.1, let $\mathbb{E}[\hat{\boldsymbol{\theta}}_2]$ be the asymptotic expectation of $\hat{\boldsymbol{\theta}}_2$.*

*We let $\boldsymbol{\theta}_1$ and $\boldsymbol{\theta}_2$ be the true parameter vectors for events 1 and 2 respectively, and assume that there exists an orthogonal matrix $\mathbf{P}$ such that $\boldsymbol{\Omega}(\boldsymbol{\theta}_1) = \mathbf{PAP}'$ and $\boldsymbol{\Omega}(\boldsymbol{\theta}_2) = \mathbf{PBP}'$ for diagonal matrices $\mathbf{A}$ and $\mathbf{B}$.*

*We then let $\mathbf{a} = \mathbf{P}\boldsymbol{\theta}_1$ and $\mathbf{b} = \mathbf{P}\mathbb{E}[\hat{\boldsymbol{\theta}}_2]$ be the projections of $\boldsymbol{\theta}_1$ and $\mathbb{E}[\hat{\boldsymbol{\theta}}_2]$ onto the column space of $\mathbf{P}$.*

*Denoting MSE as the asymptotic mean-squared error of an estimator, we find that*

$$MSE\left(\hat{\boldsymbol{\theta}}_1\right) < MSE\left(\tilde{\boldsymbol{\theta}}_1\right) \tag{9}$$

*when*

$$b_j\left(2a_j - b_j\right) > \frac{B_{j,j}}{(B_{j,j} + k)^2} \tag{10}$$

*for all $j \in \{1, p\}$.*

*The above is a sufficient, but not necessary, condition. If we denote the left-hand side of Equation 10 as $\eta_j$ and the right-hand side as $\beta_j$, and further let $\alpha_j = \frac{1}{(A_{j,j}+s)^2}$, a more relaxed condition sufficient to imply Equation 9 is that*

$$\sum_{j=1}^{p} \alpha_j \eta_j > \sum_{j=1}^{p} \alpha_j \beta_j. \tag{11}$$

*Proof.* Begin by noting that the $\text{MSE}(\hat{\boldsymbol{\theta}}_1) = \left[\text{Bias}(\hat{\boldsymbol{\theta}}_1, \boldsymbol{\theta}_1)^2 + \text{Var}(\hat{\boldsymbol{\theta}}_1)\right]$.

As shown by Phrueksawatnon et al. [2021], the asymptotic variance and squared bias of the MLE of ridge LR with ridge penalty $s$ is

$$\text{Var}(\tilde{\boldsymbol{\theta}}_1) = \text{tr}\left[\left(\boldsymbol{\Omega}(\boldsymbol{\theta}_1) + s\mathbf{I}\right)^{-1} \boldsymbol{\Omega}(\boldsymbol{\theta}_1) \left(\boldsymbol{\Omega}(\boldsymbol{\theta}_1) + s\mathbf{I}\right)^{-1}\right]$$

$$= \sum_{j=1}^{p} \frac{A_{j,j}}{(A_{j,j} + s)^2}$$

and

$$\text{Bias}(\tilde{\boldsymbol{\theta}}_1, \boldsymbol{\theta}_1)^2 = s^2 \sum_{j=1}^{p} \frac{a_j^2}{(A_{j,j} + s)^2}$$

Note that we can always diagonalize $\boldsymbol{\Omega}(\boldsymbol{\theta}_1) = \mathbf{PAP}'$ as such because it is a real symmetric matrix.

Recall that the asymptotic variance of our the similar two-step logistic regression estimator is

$$\text{Var}(\hat{\boldsymbol{\theta}}_{(s)}^r) = \left(\boldsymbol{\Omega}(\boldsymbol{\theta}_1) + s\mathbf{I}\right)^{-1} \left(\boldsymbol{\Omega}(\boldsymbol{\theta}_1) + s^2\text{Var}[\hat{\boldsymbol{\theta}}_2]\right) \left(\boldsymbol{\Omega}(\boldsymbol{\theta}_1) + s\mathbf{I}\right)^{-1}$$

Then,

$$\text{tr}\left[\left(\boldsymbol{\Omega}(\boldsymbol{\theta}_1) + s\mathbf{I}\right)^{-1} \left(\boldsymbol{\Omega}(\boldsymbol{\theta}_1) + s^2\text{Var}[\hat{\boldsymbol{\theta}}_2]\right) \left(\boldsymbol{\Omega}(\boldsymbol{\theta}_1) + s\mathbf{I}\right)^{-1}\right]$$

$$=$$

$$\text{tr}\left[\left(\boldsymbol{\Omega}(\boldsymbol{\theta}_1) + s\mathbf{I}\right)^{-1} \boldsymbol{\Omega}(\boldsymbol{\theta}_1) \left(\boldsymbol{\Omega}(\boldsymbol{\theta}_1) + s\mathbf{I}\right)^{-1}\right] + s^2\text{tr}\left[\left(\boldsymbol{\Omega}(\boldsymbol{\theta}_1) + s\mathbf{I}\right)^{-1} \text{Var}[\hat{\boldsymbol{\theta}}_2] \left(\boldsymbol{\Omega}(\boldsymbol{\theta}_1) + s\mathbf{I}\right)^{-1}\right]$$

Note the first term is equal to $\text{Var}(\tilde{\boldsymbol{\theta}}_1)$. Namely,

$$\text{tr}\left[\left(\boldsymbol{\Omega}(\boldsymbol{\theta}_1) + s\mathbf{I}\right)^{-1} \boldsymbol{\Omega}(\boldsymbol{\theta}_1) \left(\boldsymbol{\Omega}(\boldsymbol{\theta}_1) + s\mathbf{I}\right)^{-1}\right] = \text{Var}(\tilde{\boldsymbol{\theta}}_1).$$

We then expand the second term, making

$$s^2 \text{tr} \left[ \left( \mathbf{\Omega}(\boldsymbol{\theta}_1) + s\mathbf{I} \right)^{-1} \text{Var}[\hat{\boldsymbol{\theta}}_2] \left( \mathbf{\Omega}(\boldsymbol{\theta}_1) + s\mathbf{I} \right)^{-1} \right]$$

$$=$$

$$s^2 \text{tr} \left[ \left( \mathbf{\Omega}(\boldsymbol{\theta}_1) + s\mathbf{I} \right)^{-1} \left( \mathbf{\Omega}(\boldsymbol{\theta}_2) + k\mathbf{I} \right)^{-1} \mathbf{\Omega}(\boldsymbol{\theta}_2) \left( \mathbf{\Omega}(\boldsymbol{\theta}_2) + k\mathbf{I} \right)^{-1} \left( \mathbf{\Omega}(\boldsymbol{\theta}_1) + s\mathbf{I} \right)^{-1} \right]$$

where $k$ is the ridge parameter used to estimate $\hat{\boldsymbol{\theta}}_2$. Now, we use the fact that $\mathbf{\Omega}(\boldsymbol{\theta}_1) = \mathbf{PAP}'$ and $\mathbf{\Omega}(\boldsymbol{\theta}_2) = \mathbf{PBP}'$ for diagonal matrices $\mathbf{A}$ and $\mathbf{B}$. In particular, we manipulate this second term similar to Williams [2018] as shown below.

$$s^2 \text{tr} \left[ \left( \mathbf{\Omega}(\boldsymbol{\theta}_1) + s\mathbf{I} \right)^{-1} \left( \mathbf{\Omega}(\boldsymbol{\theta}_2) + k\mathbf{I} \right)^{-1} \mathbf{\Omega}(\boldsymbol{\theta}_2) \left( \mathbf{\Omega}(\boldsymbol{\theta}_2) + k\mathbf{I} \right)^{-1} \left( \mathbf{\Omega}(\boldsymbol{\theta}_1) + s\mathbf{I} \right)^{-1} \right]$$

$$=$$

$$s^2 \text{tr} \left[ \mathbf{PP}' \left( \mathbf{\Omega}(\boldsymbol{\theta}_1) + s\mathbf{I} \right)^{-1} \mathbf{PP}' \left( \mathbf{\Omega}(\boldsymbol{\theta}_2) + k\mathbf{I} \right)^{-1} \mathbf{PP}'\mathbf{\Omega}(\boldsymbol{\theta}_2)\mathbf{PP}' \left( \mathbf{\Omega}(\boldsymbol{\theta}_2) + k\mathbf{I} \right)^{-1} \mathbf{PP}' \left( \mathbf{\Omega}(\boldsymbol{\theta}_1) + s\mathbf{I} \right)^{-1} \right]$$

$$=$$

$$s^2 \text{tr} \left[ \mathbf{P}' \left( \mathbf{\Omega}(\boldsymbol{\theta}_1) + s\mathbf{I} \right)^{-1} \mathbf{PP}' \left( \mathbf{\Omega}(\boldsymbol{\theta}_2) + k\mathbf{I} \right)^{-1} \mathbf{PP}'\mathbf{\Omega}(\boldsymbol{\theta}_2)\mathbf{PP}' \left( \mathbf{\Omega}(\boldsymbol{\theta}_2) + k\mathbf{I} \right)^{-1} \mathbf{PP}' \left( \mathbf{\Omega}(\boldsymbol{\theta}_1) + s\mathbf{I} \right)^{-1} \mathbf{P} \right]$$

$$=$$

$$s^2 \text{tr} \left[ \left( \mathbf{P}'\mathbf{\Omega}(\boldsymbol{\theta}_1)\mathbf{P} + s\mathbf{I} \right)^{-1} \left( \mathbf{P}'\mathbf{\Omega}(\boldsymbol{\theta}_2)\mathbf{P} + k\mathbf{I} \right)^{-1} \mathbf{P}'\mathbf{\Omega}(\boldsymbol{\theta}_2)\mathbf{P} \left( \mathbf{P}'\mathbf{\Omega}(\boldsymbol{\theta}_2)\mathbf{P} + k\mathbf{I} \right)^{-1} \left( \mathbf{P}'\mathbf{\Omega}(\boldsymbol{\theta}_1)\mathbf{P} + s\mathbf{I} \right)^{-1} \right]$$

$$=$$

$$s^2 \text{tr} \left[ \left( \mathbf{A} + s\mathbf{I} \right)^{-1} \left( \mathbf{B} + k\mathbf{I} \right)^{-1} \mathbf{B} \left( \mathbf{B} + k\mathbf{I} \right)^{-1} \left( \mathbf{A} + s\mathbf{I} \right)^{-1} \right]$$

$$=$$

$$s^2 \sum_{j=1}^{p} \frac{B_{j,j}}{(B_{j,j} + k)^2(A_{j,j} + s)^2}.$$

Therefore, asymptotically,

$$\text{Var}(\hat{\boldsymbol{\theta}}_1) = \text{Var}(\tilde{\boldsymbol{\theta}}_1) + s^2 \sum_{j=1}^{p} \frac{B_{j,j}}{(B_{j,j} + k)^2(A_{j,j} + s)^2}.$$

We can then rewrite,

$$\text{Bias}(\hat{\boldsymbol{\theta}}_1, \boldsymbol{\theta}_1)^2 = \left[ \text{Bias}(\hat{\boldsymbol{\theta}}_1, \boldsymbol{\theta}_1) \right]' \left[ \text{Bias}(\hat{\boldsymbol{\theta}}_1, \boldsymbol{\theta}_1) \right]$$

$$= \left[ -s \left( \mathbf{\Omega}(\boldsymbol{\theta}_1) + s\mathbf{I} \right)^{-1} \left( \boldsymbol{\theta}_1 - \mathbb{E}\left[ \hat{\boldsymbol{\theta}}_2 \right] \right) \right]' \left[ -s \left( \mathbf{\Omega}(\boldsymbol{\theta}_1) + s\mathbf{I} \right)^{-1} \left( \boldsymbol{\theta}_1 - \mathbb{E}\left[ \hat{\boldsymbol{\theta}}_2 \right] \right) \right]$$

Then

$$\text{Bias}(\hat{\boldsymbol{\theta}}_1, \boldsymbol{\theta}_1)^2 = s^2 \left( \boldsymbol{\theta}_1 - \mathbb{E}\left[ \hat{\boldsymbol{\theta}}_2 \right] \right)' \left( \mathbf{\Omega}(\boldsymbol{\theta}_1) + s\mathbf{I} \right)^{-2} \left( \boldsymbol{\theta}_1 - \mathbb{E}\left[ \hat{\boldsymbol{\theta}}_2 \right] \right)$$

$$= s^2 \boldsymbol{\theta}_1' \left( \mathbf{\Omega}(\boldsymbol{\theta}_1) + s\mathbf{I} \right)^{-2} \boldsymbol{\theta}_1 - 2s^2 \boldsymbol{\theta}_1' \left( \mathbf{\Omega}(\boldsymbol{\theta}_1) + s\mathbf{I} \right)^{-2} \mathbb{E}\left[ \hat{\boldsymbol{\theta}}_2 \right] + s^2 \mathbb{E}\left[ \hat{\boldsymbol{\theta}}_2 \right]' \left( \mathbf{\Omega}(\boldsymbol{\theta}_1) + s\mathbf{I} \right)^{-2} \mathbb{E}\left[ \hat{\boldsymbol{\theta}}_2 \right]$$

Now, similar to Phrueksawatnon et al. [2021], we use the diagonalization of $\mathbf{\Omega}(\boldsymbol{\theta}_1)$ to see that

$$\text{Bias}(\hat{\boldsymbol{\theta}}_1, \boldsymbol{\theta}_1)^2 = s^2 \sum_{j=1}^{p} \frac{a_j^2}{(A_{j,j} + s)^2} - 2s^2 \sum_{j=1}^{p} \frac{a_j b_j}{(A_{j,j} + s)^2} + s^2 \sum_{j=1}^{p} \frac{b_j^2}{(A_{j,j} + s)^2}$$

where again $\mathbf{a} = \mathbf{P}\boldsymbol{\theta}_1$ and $\mathbf{b} = \mathbf{P}\mathbb{E}\left[\hat{\boldsymbol{\theta}}_2\right]$.

Now, note again that the first term above is equal to $\text{Bias}(\tilde{\boldsymbol{\theta}}_1, \boldsymbol{\theta}_1)^2$. Therefore,

$$\text{Bias}(\hat{\boldsymbol{\theta}}_1, \boldsymbol{\theta}_1)^2 = \text{Bias}(\tilde{\boldsymbol{\theta}}_1, \boldsymbol{\theta}_1)^2 - s^2 \sum_{j=1}^p \frac{1}{(A_{j,j}+s)^2} \left[2a_j b_j - b_j^2\right]$$

Putting this all together, this makes

$$\text{MSE}(\hat{\boldsymbol{\theta}}_1) = \text{MSE}(\tilde{\boldsymbol{\theta}}_1) + s^2 \sum_{j=1}^p \frac{1}{(A_{j,j}+s)^2} \frac{B_{j,j}}{(B_{j,j}+k)^2} - s^2 \sum_{j=1}^p \frac{1}{(A_{j,j}+s)^2} \left[2a_j b_j - b_j^2\right]$$

Therefore,

$$\text{MSE}\left(\hat{\boldsymbol{\theta}}_1\right) < \text{MSE}\left(\tilde{\boldsymbol{\theta}}_1\right)$$

when

$$s^2 \sum_{j=1}^p \frac{1}{(A_{j,j}+s)^2} \left[2a_j b_j - b_j^2\right] > s^2 \sum_{j=1}^p \frac{1}{(A_{j,j}+s)^2} \frac{B_{j,j}}{(B_{j,j}+k)^2}.$$

Which of course holds under the weaker condition that

$$b_j \left(2a_j - b_j\right) > \frac{B_{j,j}}{(B_{j,j}+k)^2}$$

for all $j \in \{1, p\}$.

$\square$

**Note on implications of Theorem 4.2**   Because $\forall j, \beta_j > 0$, it is necessary for $|a_j| > |a_j - b_j|$ in order for the inequality in Equation 10 to hold. If we further observe that $\text{Var}(\hat{\boldsymbol{\theta}}_2) = \sum_{j=1}^p \beta_j$, Theorem 4.2 shows that the degree to which $\mathbf{a}$ is closer to $\mathbf{b}$ than $\mathbf{0}$ must be enough to account for the added variance of estimating $\boldsymbol{\theta}_2$. And noting that $\mathbf{a} = \mathbf{P}\boldsymbol{\theta}_1$ and $\mathbf{b} = \mathbf{P}\mathbb{E}[\hat{\boldsymbol{\theta}}_2]$, the most intuitive way for $\mathbf{a}$ to be close to $\mathbf{b}$ is for $\boldsymbol{\theta}_1$ to be close to $\mathbb{E}[\hat{\boldsymbol{\theta}}_2]$.

## A.1   ASYMPTOTIC PROPERTIES OF CET-LR

**Theorem A.1.** *Let $\mathcal{L}^{(s,k)}(\boldsymbol{\theta}|\mathcal{D}_n) = \mathcal{L}^{(s)}(\boldsymbol{\theta}|\mathcal{D}_n) - \frac{1}{2}k\|\boldsymbol{\theta}\|_2^2$ be the log-likelihood of CET-LR with an added $L_2$ regularization penalty on the magnitude of $\boldsymbol{\theta}$. Then, the MLE estimate of $\mathcal{L}^{(s,k)}(\boldsymbol{\theta}|\mathcal{D}_n)$, denoted $\hat{\boldsymbol{\theta}}^{(s,k)}$, has asymptotic bias*

$$\mathbb{E}[\hat{\boldsymbol{\theta}}^{(s,k)} - \boldsymbol{\theta}] = -\left\{\boldsymbol{\Omega}^{(s,k)}(\boldsymbol{\theta})\right\}^{-1} \left(k\boldsymbol{\theta} + s\begin{bmatrix}\boldsymbol{\theta}_1 - \boldsymbol{\theta}_2 \\ \boldsymbol{\theta}_2 - \boldsymbol{\theta}_1\end{bmatrix}\right) \tag{13}$$

*and asymptotic variance*

$$\text{Var}(\hat{\boldsymbol{\theta}}^{(s,k)}) = \left\{\boldsymbol{\Omega}^{(s,k)}(\boldsymbol{\theta})\right\}^{-1} \boldsymbol{\Omega}(\boldsymbol{\theta})\left\{\boldsymbol{\Omega}^{(s,k)}(\boldsymbol{\theta})\right\}^{-1}. \tag{14}$$

*Above, we recall that*

$$\boldsymbol{\theta} = \begin{bmatrix}\boldsymbol{\theta}_1 \\ \boldsymbol{\theta}_2\end{bmatrix}$$

*and use $\boldsymbol{\Omega}(\boldsymbol{\theta})$ and $\boldsymbol{\Omega}^{(s,k)}(\boldsymbol{\theta})$ to denote the negative of the hessian matrix from the unregularized log-likelihood and $\mathcal{L}^{(s,k)}(\boldsymbol{\theta}|\mathcal{D}_n)$ respectively.*

*Proof.* We proceed similarly to the proof for Theorem 4.1. Thus,

$$\mathbf{U}^{(s,k)}(\boldsymbol{\theta}) = \mathbf{U}(\boldsymbol{\theta}) - k\boldsymbol{\theta} - s \begin{bmatrix} \boldsymbol{\theta}_1 - \boldsymbol{\theta}_2 \\ \boldsymbol{\theta}_2 - \boldsymbol{\theta}_1 \end{bmatrix}$$

and

$$\boldsymbol{\Omega}^{(s,k)}(\boldsymbol{\theta}) = \boldsymbol{\Omega}(\boldsymbol{\theta}) + k\mathbf{I}^* + s \left( \mathbf{I}^* + \begin{bmatrix} \mathbf{0} & -\mathbf{I} \\ -\mathbf{I} & \mathbf{0} \end{bmatrix} \right).$$

$\mathbf{I}^*$ is a $2p \times 2p$ identity matrix and

$$\begin{bmatrix} \mathbf{0} & -\mathbf{I} \\ -\mathbf{I} & \mathbf{0} \end{bmatrix}$$

is a $2p \times 2p$ matrix where $\mathbf{I}$ is a $p \times p$ identity matrix and $\mathbf{0}$ is a $p \times p$ matrix of all zeros.

Then, we derive the large sample properties of our estimate, $\hat{\boldsymbol{\theta}}^{(s,k)}$, with the Taylor series expansion about the true parameter $\boldsymbol{\theta}$. This gives

$$\mathbf{U}^{(s,k)}(\hat{\boldsymbol{\theta}}^{(s,k)}) = \mathbf{U}^{(s,k)}(\boldsymbol{\theta}) - \left( \hat{\boldsymbol{\theta}}^{(s,k)} - \boldsymbol{\theta} \right) \boldsymbol{\Omega}^{(s,k)}(\boldsymbol{\theta}) + o\left( \|\hat{\boldsymbol{\theta}}^{(s,k)} - \boldsymbol{\theta}\| \right)$$

and the first-order approximation is

$$\hat{\boldsymbol{\theta}}^{(s,k)} = \boldsymbol{\theta} + \left\{ \boldsymbol{\Omega}^{(s,k)}(\boldsymbol{\theta}) \right\}^{-1} \left( \mathbf{U}(\boldsymbol{\theta}) - k\boldsymbol{\theta} - s \begin{bmatrix} \boldsymbol{\theta}_1 - \boldsymbol{\theta}_2 \\ \boldsymbol{\theta}_2 - \boldsymbol{\theta}_1 \end{bmatrix} \right).$$

From here, we arrive at an asymptotic bias

$$\mathbb{E}[\hat{\boldsymbol{\theta}}^{(s,k)} - \boldsymbol{\theta}] = - \left\{ \boldsymbol{\Omega}^{(s,k)}(\boldsymbol{\theta}) \right\}^{-1} \left( k\boldsymbol{\theta} + s \begin{bmatrix} \boldsymbol{\theta}_1 - \boldsymbol{\theta}_2 \\ \boldsymbol{\theta}_2 - \boldsymbol{\theta}_1 \end{bmatrix} \right).$$

Now, noting as in Theorem 4.1 that the unregularized MLE estimate $\hat{\boldsymbol{\theta}} = \boldsymbol{\theta} + \boldsymbol{\Omega}^{-1}(\boldsymbol{\theta})\mathbf{U}(\boldsymbol{\theta})$, we rewrite

$$\begin{aligned}
\hat{\boldsymbol{\theta}}^{(s,k)} &= \left\{ \boldsymbol{\Omega}^{(s,k)}(\boldsymbol{\theta}) \right\}^{-1} (\mathbf{U}(\boldsymbol{\theta}) - \boldsymbol{\Omega}(\boldsymbol{\theta})\boldsymbol{\theta}) \\
&= \left\{ \boldsymbol{\Omega}^{(s,k)}(\boldsymbol{\theta}) \right\}^{-1} \left( \mathbf{U}(\boldsymbol{\theta}) - \boldsymbol{\Omega}(\boldsymbol{\theta}) \left( \hat{\boldsymbol{\theta}} - \boldsymbol{\Omega}^{-1}(\boldsymbol{\theta})\mathbf{U}(\boldsymbol{\theta}) \right) \right) \\
&= \left\{ \boldsymbol{\Omega}^{(s,k)}(\boldsymbol{\theta}) \right\}^{-1} \boldsymbol{\Omega}(\boldsymbol{\theta})\hat{\boldsymbol{\theta}}
\end{aligned}$$

.

And since the asymptotic variance of $\hat{\boldsymbol{\theta}} = \boldsymbol{\Omega}^{-1}(\boldsymbol{\theta})$, we have the asymptotic variance

$$\mathrm{Var}(\hat{\boldsymbol{\theta}}^{(s,k)}) = \left\{ \boldsymbol{\Omega}^{(s,k)}(\boldsymbol{\theta}) \right\}^{-1} \boldsymbol{\Omega}(\boldsymbol{\theta}) \left\{ \boldsymbol{\Omega}^{(s,k)}(\boldsymbol{\theta}) \right\}^{-1}.$$

$\qquad\square$

We establish Lemma A.2 for use in our proof to Theorem 4.3.

**Lemma A.2.** *Let $\mathbf{A}$ be an $n \times n$ diagonalizable matrix such that $\mathbf{A} = \mathbf{Q}\boldsymbol{\Lambda}\mathbf{Q}^{-1}$ where $\{\lambda_i\}_{i=1}^n$ are the $n$ diagonal entries of $\boldsymbol{\Lambda}$ and eigenvalues of $\mathbf{A}$. Then for an $n \times n$ identity matrix $\mathbf{I}$ and any real constant $c$ we have that the eigenvalues of the $2n \times 2n$ matrix*

$$\mathbf{B} = \begin{bmatrix} \mathbf{A} + c\mathbf{I} & -c\mathbf{I} \\ -c\mathbf{I} & \mathbf{A} + c\mathbf{I} \end{bmatrix}$$

*are $\{\lambda_i\}_{i=1}^n \cup \{\lambda_i + 2c\}_{i=1}^n$.*

*Proof.* We find the eigenvalues by solving the equation

$$\det(\mathbf{B} - \lambda\mathbf{I}^*) = 0$$

where $\mathbf{I}^*$ is a $2n \times 2n$ matrix. Note that

$$\mathbf{B} - \lambda\mathbf{I}^* = \begin{bmatrix} \mathbf{A} + (c - \lambda)\mathbf{I} & -c\mathbf{I} \\ -c\mathbf{I} & \mathbf{A} + (c - \lambda)\mathbf{I} \end{bmatrix}$$

Since $\mathbf{A} + (c - \lambda)\mathbf{I}$ and $-c\mathbf{I}$ commute with each other, by Silvester [2000] we have that

$$
\begin{aligned}
\det(\mathbf{B} - \lambda\mathbf{I}^*) &= \det\left((\mathbf{A} + (c - \lambda)\mathbf{I})^2 - (-c\mathbf{I})^2\right) \\
&= \det\left(\mathbf{A} + (c - \lambda)\mathbf{I} - c\mathbf{I}\right) \\
&\quad \det\left(\mathbf{A} + (c - \lambda)\mathbf{I} + c\mathbf{I}\right) \\
&= \det\left(\mathbf{A} - \lambda\mathbf{I}\right)\det\left(\mathbf{A} + (2c - \lambda)\mathbf{I}\right)
\end{aligned}
\tag{15}
$$

Therefore, setting $\det(\mathbf{B} - \lambda\mathbf{I}^*) = 0$, and noting that the eigenvalues of $\mathbf{A}$ are $\{\lambda_i\}_{i=1}^n$, we conclude that the eigenvalues of $\mathbf{B}$ are $\{\lambda_i\}_{i=1}^n \cup \{\lambda_i + 2c\}_{i=1}^n$. $\qquad\square$

**Theorem 4.3** (CET-LR Asymptotic MSE). *If $\boldsymbol{\theta}_1 = \boldsymbol{\theta}_2$, then for any $s' > s \geq 0$ the asymptotic MSE of the MLE estimate of $\boldsymbol{\theta} = [\boldsymbol{\theta}_1, \boldsymbol{\theta}_2]$ is less under the log-likelihood of $\mathcal{L}^{(s')}(\boldsymbol{\theta}|\mathcal{D}_n)$ than $\mathcal{L}^{(s)}(\boldsymbol{\theta}|\mathcal{D}_n)$.*

*Proof.* Start by noting that this log-likelihood is of the same form as $\mathcal{L}^{(s,k)}$ from Theorem A.1 with $k = 0$. Therefore, we use those same results, setting $k = 0$ and dropping $k$ from the superscript. Then, with $\boldsymbol{\theta}_1 = \boldsymbol{\theta}_2$, Theorem A.1 shows that $\mathcal{L}^{(s)}$ is asymptotically unbiased for any value of $s$.

Therefore, the asymptotic MSE is just the asymptotic variance of the estimate, $\hat{\boldsymbol{\theta}}^{(s)}$.

$$
\mathrm{Var}(\hat{\boldsymbol{\theta}}^{(s)}) = \mathrm{tr}\left[\left\{\boldsymbol{\Omega}^{(s)}(\boldsymbol{\theta})\right\}^{-1}\boldsymbol{\Omega}(\boldsymbol{\theta})\left\{\boldsymbol{\Omega}^{(s)}(\boldsymbol{\theta})\right\}^{-1}\right]
$$

We observe that $\boldsymbol{\Omega}^{(s)}(\boldsymbol{\theta})$ has the following structure

$$
\begin{bmatrix}
\boldsymbol{\Omega}(\boldsymbol{\theta}_1) + s\mathbf{I} & -s\mathbf{I} \\
-s\mathbf{I} & \boldsymbol{\Omega}(\boldsymbol{\theta}_1) + s\mathbf{I}
\end{bmatrix}
$$

From here, using Lemma A.2 we have that

$$
\mathrm{MSE}(\hat{\boldsymbol{\theta}}^{(s)}) = \mathrm{Var}(\hat{\boldsymbol{\theta}}^{(s)}) = \sum_{j=1}^p \frac{1}{\lambda_j} + \frac{\lambda_j}{(\lambda_j + s)^2}
$$

where $\lambda_j$ is the $j$-th eigenvalue of $\boldsymbol{\Omega}(\boldsymbol{\theta}_1)$.

Therefore, for any $s' > s \geq 0$, we have that

$$
\mathrm{MSE}(\hat{\boldsymbol{\theta}}^{(s')}) < \mathrm{MSE}(\hat{\boldsymbol{\theta}}^{(s)}).
$$

And we note that as $s \to \infty$, $\mathrm{MSE}(\hat{\boldsymbol{\theta}}^{(s)}) \to \sum_{j=1}^p \frac{1}{\lambda_j}$ which is the MSE of the MLE estimator of the unregularized log-likelihood of $\boldsymbol{\theta}_1$.

$\qquad\square$

# B EXPERIMENTAL DETAILS

## B.1 DATA GENERATING PROCESS (DGP) FOR SYNTHETIC DATASET

In simulation experiments, we first generated a synthetic dataset $\mathbf{X} \in \mathbb{R}^{n \times k}$ comprising $n$ patients with $k$ patient features. All patient features are independently and identically sampled from a normal distribution of $\mathcal{N}(0, 10)$. We then use linear or non-linear DGP to generate observations of two binary outcomes, $y_{i,1}$ and $y_{i,2}$, with expected event rate $\pi_{i,1}$ and $\pi_{i,2}$, and expected event similarity $\rho$ between $y_{i,1}$ and $y_{i,2}$. Only a randomly-selected subset of the feature matrix, $\mathbf{X}^{(r)} \in \mathbb{R}^{n \times r}$ with $r \leq k$ relevant features, are used to compute outcome probabilities, while the remaining features served as noise. We set $k = 25$ and $r = 20$ for all our simulations.

For the linear DGP, we first generated a pair of standardized $r$-dimension vectors $\boldsymbol{\theta}_1$ and $\boldsymbol{\theta}_2$ with cosine similarity $\rho$, and used these vectors as feature coefficients to compute initial logits for $y_{i,1}$ and $y_{i,2}$ through linear combinations with $\mathbf{x}_i^{(r)}$. The intercept terms $\gamma_1$ and $\gamma_2$ are searched as offsets to align event probability with expected event rates $\pi_1$ and $\pi_2$, which can be written as

$$P(y_{i,1} = 1 | \mathbf{x}_i) = \sigma(\boldsymbol{\theta}_1 \mathbf{x}_i^{(r)} + \gamma_1), \text{ and}$$
$$P(y_{i,2} = 1 | \mathbf{x}_i) = \sigma(\boldsymbol{\theta}_2 \mathbf{x}_i^{(r)} + \gamma_2), \tag{16}$$

where $\sigma$ denotes the sigmoid function. Finally, the synthetic observations of $y_{i,1}$ and $y_{i,2}$ are generated through a random Bernoulli draw based on the event probability for each patient.

For the non-linear DGP, we used two mapping function $h_1(\cdot)$ and $h_2(\cdot)$ that map $\mathbf{x}_i^{(r)}$ to $l$-dimensional latent feature vectors $\mathbf{x}_{i,1}^{(l)}$ and $\mathbf{x}_{i,2}^{(l)}$. Specifically, we generated two orthogonal mapping matrices, $\mathbf{W}_1$ and $\mathbf{W}_2$, each combined with a ReLU activation function. Thus the mapping can be written as

$$\mathbf{x}_{i,1}^{(l)} = h_1(\mathbf{x}_i^{(r)}) = \text{ReLU}(\mathbf{W}_1 \mathbf{x}_i^{(r)}), \text{ and}$$
$$\mathbf{x}_{i,2}^{(l)} = h_2(\mathbf{x}_i^{(r)}) = \text{ReLU}(\mathbf{W}_2 \mathbf{x}_i^{(r)}), \tag{17}$$

where $\mathbf{W}_1 \in \mathbb{R}^{l \times r}$, $\mathbf{W}_2 \in \mathbb{R}^{l \times r}$. We then follow the same process to generate outcome observations as linear DGP but replace the features vector $\mathbf{x}_i^{(r)}$ with latent feature vector $\mathbf{x}_i^{(l)}$. The event probabilities in non-linear DGP would subsequently become

$$P(y_{i,1} = 1 | \mathbf{x}_i) = \sigma(\boldsymbol{\theta}_1 \mathbf{x}_{i,1}^{(l)} + \gamma_1), \text{ and}$$
$$P(y_{i,2} = 1 | \mathbf{x}_i) = \sigma(\boldsymbol{\theta}_2 \mathbf{x}_{i,2}^{(l)} + \gamma_2). \tag{18}$$

Specifically, the non-linear DGP partially shares $l \times \rho$ latent features between $\mathbf{x}_{i,1}^{(l)}$ and $\mathbf{x}_{i,2}^{(l)}$. In other words, a subset proportion of $\rho$ is used to select subsets from mapping matrices $\mathbf{W}_1$ and $\mathbf{W}_2$, as well as the corresponding subsets from the coefficient vectors $\boldsymbol{\theta}_1$ and $\boldsymbol{\theta}_2$, which are designed to be identical.

The sample size of the synthetic datasets are $n = 50,000$ for linear DGP experiments, and increase to $n = 250,000$ for non-linear DGP experiments. The size of latent space for non-linear DGP is set as $l = 5$ in our simulation experiments.

In experiments varying event similarity or event rate, we only modify the synthetic observations of common outcomes $y_2$, while maintaining the feature matrix $\mathbf{X}$ and observations of rare outcome $y_1$ consistent across different experiment setups with same random seed.

## B.2 ADDITIONAL DETAILS FOR MODEL TRAINING

In both simulation and real-world experiments, we conduct 10 iterations under every experimental setup. For each iteration, we either randomly generate a synthetic dataset, or conduct a random partitioning to generate the training and testing sets on the real-world datasets. The random seeds are always set to match with the iteration number.

We allocate 25% of the samples from the training set for validation. In the validation for MLL, simulation experiments use aggregated performance across all outcomes as the criterion, whereas real-world experiments focus solely on the outcome of interest. The learning rate, batch size, and hidden layer size (exclusively for NN models), are pre-tuned and fixed as constant across iterations. The strength parameters for ridge regularization and similarity penalty were dynamically learned for each iterations by grid search based on the validation performance of AUC. Early stopping are also implemented to avoid overfitting based on validation performance of the cross-entropy loss.

# C ADDITIONAL EXPERIMENTAL RESULTS

## C.1 COMPARISON WITH ALTERNATIVE APPROACHES

We compare our proposed methods with several established approaches in the context of rare event prediction and multi-event information sharing. For single-label learning, we use Firth logistic regression and gradient boosting as two alternative baselines, known for their high performance in rare event prediction with high-dimensional datasets similar to our setup [Doerken et al., 2019]. We conduct feature selection using LASSO prior to running Firth logistic regression to avoid convergence issues that arise on large datasets with linearly dependent variables. For multi-event learning, we include transfer learning as an additional baseline where neural networks were pre-trained on common events and then fine tuned on the target event.

The rare event prediction performance of all alternative approaches, as well as the methods we propose, is summarized in Table 1 and Table 2 for synthetic datasets using non-linear DGP and real-world datasets, respectively. For CET methods, we only include the result of using $L_2$ magnitude for similarity.

Notably, our proposed method CET-NN outperforms all other methods across all real-world datasets and in the synthetic dataset when event similarity exceeds 40%.

Table 1: Average (standard deviation) AUC or Spearmen's correlation coefficient on synthetic non-linear DGP.

| | | Multi-label learning | | | Single-label learning | | | |
|---|---|---|---|---|---|---|---|---|
| | Similarity | Multilabel NN | Transfer NN | CET NN | LR | Firth LR | GDBoost | NN |
| AUC | 100% | 0.880 (.070) | 0.886 (.067) | 0.895 (.067) | | | | |
| | 80% | 0.880 (.069) | 0.883 (.070) | 0.890 (.070) | | | | |
| | 60% | 0.878 (.071) | 0.878 (.071) | 0.884 (.072) | 0.884 | 0.884 | 0.869 | 0.871 |
| | 40% | 0.875 (.072) | 0.877 (.074) | 0.881 (.073) | (.079) | (.078) | (.081) | (.074) |
| | 20% | 0.874 (.072) | 0.875 (.072) | 0.876 (.076) | | | | |
| | 0% | 0.873 (.073) | 0.874 (.075) | 0.876 (.073) | | | | |
| $\rho$ | 100% | 0.842 (.045) | 0.861 (.050) | 0.900 (.040) | | | | |
| | 80% | 0.834 (.049) | 0.849 (.048) | 0.873 (.042) | | | | |
| | 60% | 0.820 (.047) | 0.828 (.046) | 0.840 (.042) | 0.778 | 0.780 | 0.731 | 0.796 |
| | 40% | 0.811 (.051) | 0.826 (.047) | 0.824 (.051) | (.049) | (.049) | (.049) | (.051) |
| | 20% | 0.801 (.051) | 0.806 (.045) | 0.796 (.053) | | | | |
| | 0% | 0.799 (.055) | 0.799 (.045) | 0.797 (.049) | | | | |

Table 2: Average (standard deviation) AUC on real-world rare disease.

| Target | Secondary disease | Multi-label learning | | | Single-label learning | | | |
|---|---|---|---|---|---|---|---|---|
| | | Multilabel NN | Transfer NN | CET NN | LR | Firth LR | GDBoost | NN |
| Stroke | Hypertensive crisis | 0.652 (.008) | 0.639 (.012) | 0.670 (.017) | 0.605 | 0.633 | 0.643 | 0.627 |
| Stroke | Heart failure | 0.651 (.010) | 0.636 (.020) | 0.656 (.013) | (.018) | (.019) | (.016) | (.009) |
| Stroke | Renal failure | 0.653 (.014) | 0.630 (.014) | 0.656 (.025) | | | | |
| Autism | Any other ND | 0.770 (.019) | 0.731(.026) | 0.775 (.020) | 0.721 | 0.733 | 0.726 | 0.738 |
| Autism | Language delay | 0.768 (.021) | 0.743 (.024) | 0.772 (.025) | (.022) | (.029) | (.020) | (.023) |
| Autism | Motor delay | 0.737 (.014) | 0.732 (.025) | 0.742 (.017) | | | | |

## C.2 ADDITIONAL RESULTS FOR SIMULATIONS VARYING SIMILARITY

In the simulation experiments using the linear DGP, we test the performance of CET-LR for both rare and common outcomes with two evaluation metrics, Spearman's rank correlation $\rho$ and AUC. The performance of AUC in Figure 5b shows consistent trend with $\rho$ in Figure 5a, supporting our strategy of using AUC as a proxy for real-world dataset evaluation. Figure 5c shows that the performance for $y_2$ is mostly consistent with the baseline, indicating that the learning for the common outcome is not be compromised by the CET approach.

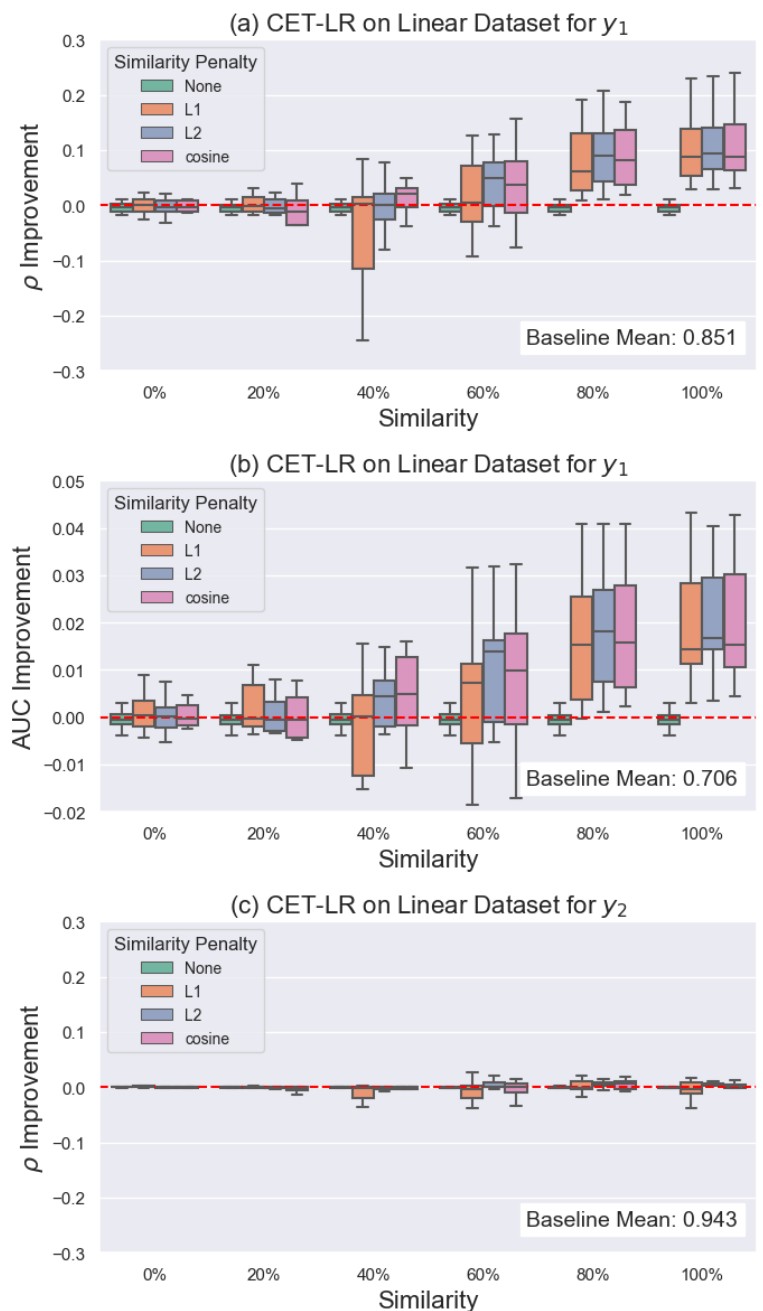

Figure 5: Boxplots representing pairwise enhancement of (a) Spearman's rank ($\rho$) for the rare outcome $y_1$, (b) AUC for the rare outcome $y_1$, (c) Spearman's rank for the common outcome $y_2$. The red line indicates baseline of single-label learning for each setup across iterations.

Besides the experiments for models on the corresponded synthetic datasets, i.e., LR for linear DGP and NN for non-linear DGP, we also investigate the LR model performance on the non-linear synthetic dataset. Figrue 6 shows that the mismatch between model structure and underlying generative function not only leads to a decreased performance, but further invalidate the CET enhancement across all levels of event similarity.

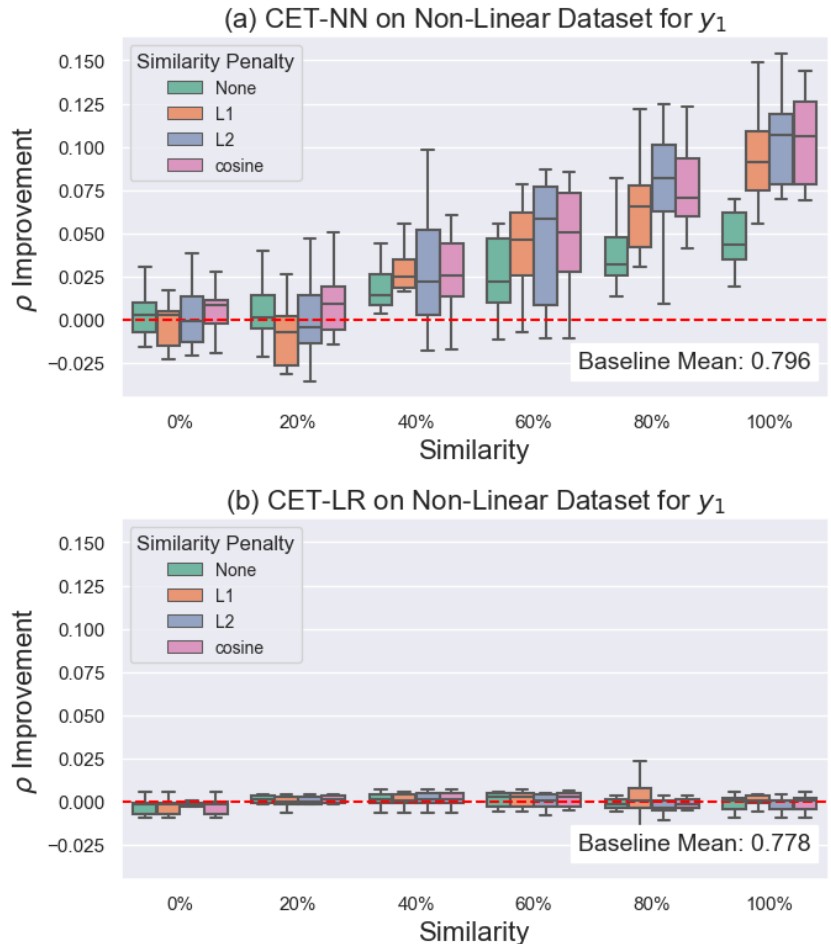

Figure 6: Boxplots representing pairwise enhancement for the rare outcome $y_1$ of (a) CET-LR, (b) CET-NN on non-linear DGP generated datasets. The red line indicates baseline of single-label learning for each iteration.

### C.3 ADDITIONAL RESULTS FOR SIMILARITY PENALTY STRENGTH

To support the claim that the performance enhancement for CET methods is from the added similarity term, we plot the similarity penalty parameter ($s$) selected via validation versus the underlying event similarity. Figure 7 shows the positive correlation between these two factors. It is worth mentioning that this behavior makes our method robust to imposing a similarity penalty on unrelated events when a validation set is used to tune $s$. The fact that this behavior is more apparent in CET-LR than CET-NN may be due to our earlier observation that MLL using NNs can leverage shared information across events even without a similarity penalty.

### C.4 ADDITIONAL RESULTS FOR SIMULATION VARYING EVENT RATE

Figure 8 shows that increasing the common event rate has a similar impact on prediction performance for CET-LR in a linear setting (Figure 2) and CET-NN in a non-linear setting (Figure 8). Both patterns support the claim that more common events can help rarer event prediction by leveraging additional information via CET methods.

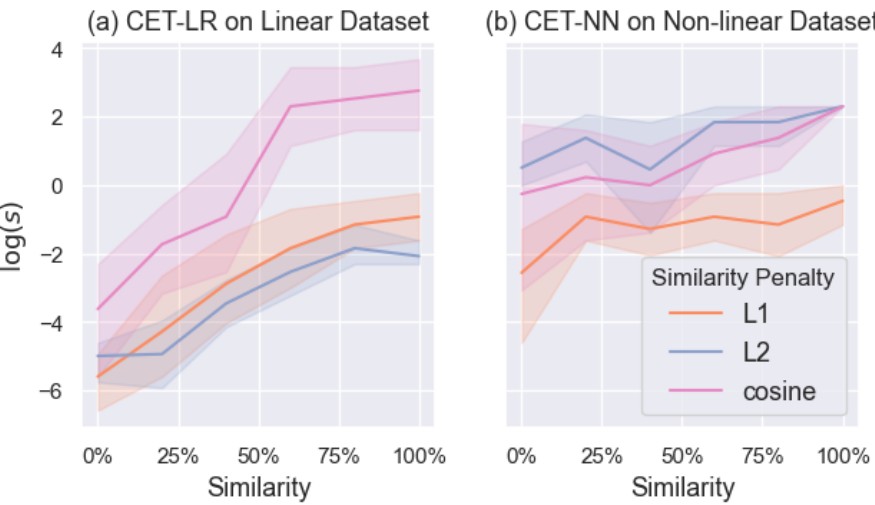

Figure 7: Log score of similarity penalty parameter $s$ learned by validation. Shaded areas represent 95% confidence intervals.

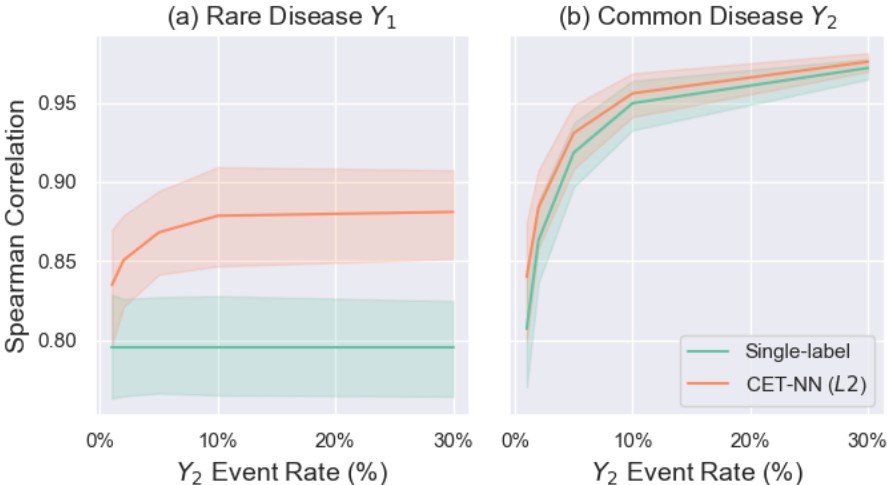

Figure 8: Performance of single-label learning and CET-NN on (a) rare and (b) common diseases generated by the non-linear DGP. The rare disease ($y_1$) event rate is 1%, and the common disease ($y_2$) event rate is varied from 1% to 30%. Shading represents 95% confidence intervals.

## C.5 ADDITIONAL RESULTS FOR LR MODELS ON REAL-WORLD DATASETS

In this section, we show results using LR models on our real-world datasets. We replot the results using the NN models for the sake of comparison (these are the results shown in Section 6 of the main text). Figure 9 shows that both CET-LR and CET-NN significantly enhance the stroke prediction when incorporating a more common outcome, implying a substantial similarity in patient features or latent features across various maternal morbidities. It is notable that models with an NN structure achieve superior baseline performance compared to those using LR in both real-world experiments of stroke (Figure 9) and autism prediction (Figure 10), and show a more pronounced enhancement effect through CET. The performance gap between LR and NN is especially significant in the autism dataset, which suggests the embedding features derived from diagnosis and procedures in EHRs are nonlinearly associated with ND outcomes.

We additionally explore the setting of utilizing multiple common events for CET approach in the preeclampsia study, which includes all pairs of coefficients to the CET penalty (Figure 9). The result shows the performance without CET penalty significantly enhanced, but minimal additional improvement from CET penalty are observed .

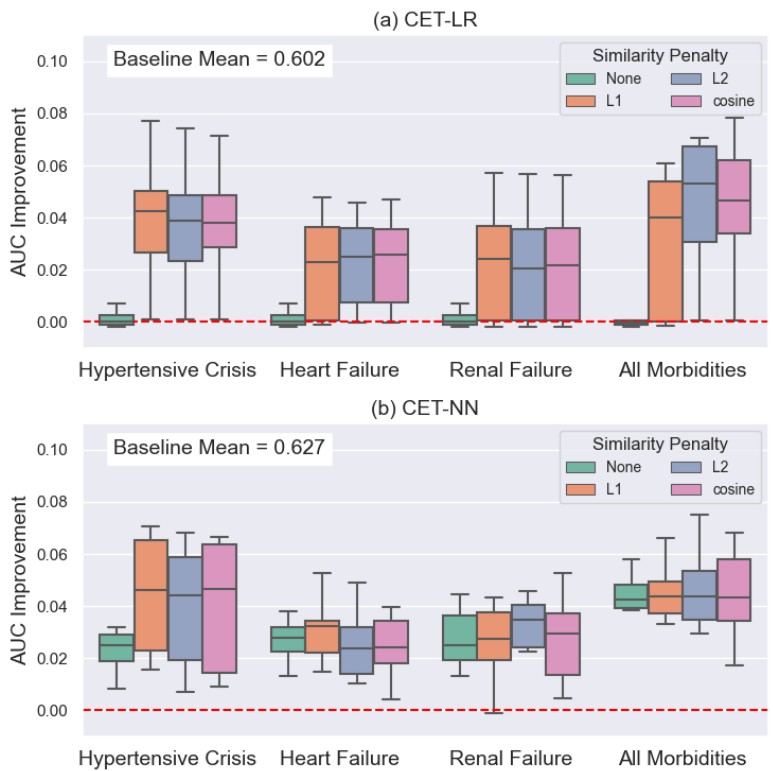

Figure 9: Boxplots showing pairwise improvement in the AUC of stroke prediction via (a) CET-LR, (b) CET-NN across 10 times resampling. The red line indicates the single-label learning baseline.

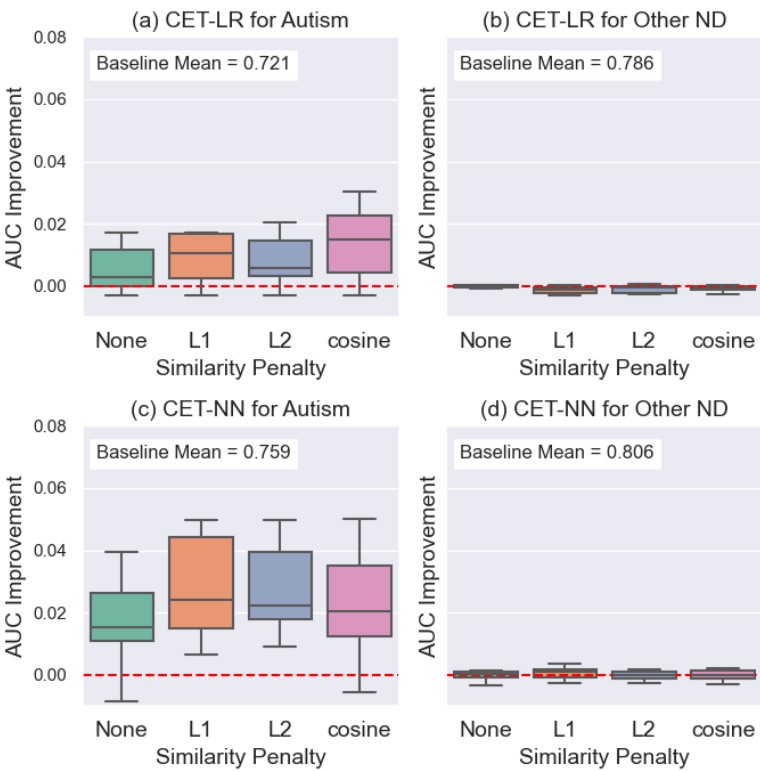

Figure 10: Boxplots showing pairwise performance improvement in the AUC of (a) autism via CET-LR, (b) other neurodevelopmental diagnoses (ND) via CET-LR, (c) autism via CET-NN, (d) other ND via CET-NN across 10 times resampling. The red line indicates the single-label learning baseline.

## C.6    ADDITIONAL RESULTS ON EXPANDED TRAINING SET FOR PREECLAMPSIA STUDY

We performed an alternative partitioning by enlarging the training set to 270,000 samples to test the method efficacy in datasets with enriched sample size. This results in an improved baseline performance, yet a diminished enhancement from MLL, shows in Figure 11.

## C.7    ADDITIONAL RESULTS OF USING SINGLE ND EVENT FOR AUTISM STUDY

As discussed in Section 7, including additional outcomes could benefit CET by increasing the common event rate. However, there is also the potential that including more unrelated events could reduce the similarity between events. In our autism study in Section 6, we defined our secondary outcome as a union over several events. This can be seen as a simplistic approach for utilizing more than two outcomes. To investigate the trade-off of including more events in our common event, we examined the performance of MLL and CET methods on the autism dataset when using a single ND outcome as the common outcome or combining multiple NDs into one common outcome (event rate 18.5%). Specifically, we considered using the two most prevalent ND events, language delay (15.6%) and motor delay (5.7%), as the common outcome. The results in Table 2 show that the combined common event demonstrates superior performance than either of these single ND events. Additionally, the CET method is significantly better when using language delay (AUC: 0.772) compared to motor delay (AUC: 0.742), which further validates the enhanced effectiveness of using events with a higher event rate as surrogate events.

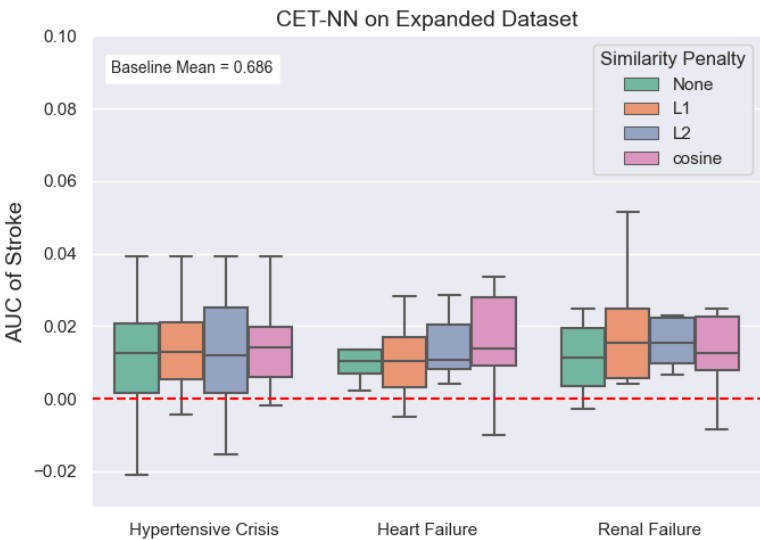

Figure 11: Boxplots showing pairwise improvement in the AUC of stroke prediction via CET-NN on the expanded dataset across 10 times resampling. The red line indicates the single-label learning baseline.