# OpenReview forum: "Common Event Tethering to Improve Prediction of Rare Clinical Events"
_auai.org/UAI/2024/Conference — UAI 2024 spotlight_

### Official Review · Reviewer_6sEz · 2024-03-14

**Q2-1 Originality-Novelty:** 3
**Q2-2 Correctness-Technical Quality:** 3
**Q2-5 Clarity Of Writing:** 4

**Q1 Summary And Contributions:**

The paper explores how the framework of multi label learning (MLL) can be use for rare event prediction. Specifically the paper leverages multiple streams of data with high correlation to gain predictive power from one high frequency event stream to a low frequency event stream. The paper provides a theoretical framework and empirical results.

**Q2-3 Extent To Which Claims Are Supported By Evidence:**

3: Good: the main claims are supported by convincing evidence (in the form of adequate experimental evaluation, proofs, (pseudo-)code, references, assumptions).

**Q2-4 Reproducibility:**

4: Excellent: key resources (e.g. proofs, code, data) are available and key details (e.g. proof sketches, experimental setup) are comprehensively described for competent researchers to confidently and easily reproduce the main results.

**Q3 Main Strengths:**

- Very well written and easy to follow
- Clearly explore novel statistical ML domain with well motivated theoretical and empirical methods. Provide guidance under what conditions their new approach is useful both theoretically and using numeric simulations. The subsequent real world experiment used complex and varied data that provide insight into the method, rather than simple toy datasets
- Extend their theoretically grounded method to a deep learning approach which can help with practical use of the approach
- Approach seems novel and a clear departure from previous literature (though I'm not an expert in MLL so i mostly judge based on the background and literature in the paper)

**Q4 Main Weakness:**

- The paper limited itself to considering settings where M=2. While the authors acknowledge that it's possible to consider a more general case of M>=2 in Section 2, that is not explored in the paper. That seems like an unfortunate and artificial limitation both theoretically and practically. In particular, the healthcare real world data that they explore actually has multiple data streams. And while the paper explores pairwise combinations, it seems more natural to include all of them in an MLL setting. Indeed, that would seem to provide more statistical power.
- Within the numeric and real world results, the authors often compared between MLL and the baseline. Implicit in this comparison is that the MLL case of "no penalty", when s=0, is considered to be a MLL case. However, when s=0 that coresponds to preexisting MLL models, meaning MLL models that do not have the specific regularization proposed by this paper. If instead the authors distinguished between the baseline MLL model and their novel MLL model, the results are substantially less impressive:
-- The numeric results then show that their regularization only is marginally more effective at a similarity score of 80% of larger.
-- The real world results then show that only in the case of Hypertension does their proposed method provide marginal advantage
-In Section 6.1 it's odd that the authors only trained on 1/6 of the data. The reasoning that they wanted enough data for testing is a bit odd since yI would think that if you have issues with statical computation of testing error, then you would have even greater difficulty with training. This isn't necessarily bad but seems like there might be something else happening there.

**Q5 Detailed Comments To The Authors:**

See above

**Q9 Complying With Reviewing Instructions:**

Yes

---

> ### Author Rebuttal · Authors · 2024-04-07
>
> We thank you for your detailed response and appreciate the opportunity to address your concerns. We organized our response into three sections that cover *experimental results*, *restriction to M=2*, and the *splitting strategy in Section 6.1*.
> - *Experimental results*: You are correct that when s=0 the comparison method corresponds to preexisting MLL methods. We will make this clearer in our figure descriptions. In response to your point that our results are substantially less impressive when compared to this baseline, we argue that our simulation results do an effective job of showing the potential advantages while acknowledging the limitations of the proposed penalty. The idea of regularized MLL has been around for a while and, to our knowledge, our work is the first exploration of the conditions under which the added penalty term is beneficial. We effectively show that when events are very similar, there is a performance gain from tethering to the more common event, but that these gains quickly diminish as the events get less similar. Importantly, our method does not hurt performance in the case that the more common surrogate event is not chosen perfectly, and small gains in performance can be meaningful when predicting life-altering medical events.
> As mentioned in our response to reviewer W1Nx, a natural follow-up question is why we did not consider more real-world examples where the surrogate event is chosen well. In response, we note that while the problem characteristics motivating this work are common in healthcare settings, the corresponding datasets are difficult or expensive to obtain due to privacy and other concerns. Because of this, our experimental results repurposed existing datasets to fit our use case and were limited by the existing outcomes to show the potential gains and limitations of our method. Our results establish the potential benefit of building datasets that include more common events that are not of primary concern but associated with a rarer event of interest for improved prediction via ours and other methods.
> - *Restriction to M=2*: We focus on M=2 in our theory and simulation studies to simplify the exploration of how event similarity and event rate impact the effectiveness of our proposed approach. As we noted above, this was an important and distinctive aspect of our paper's contribution that is not present in previous literature.
> The primary reason we continued to use M=2 in our real-world experimental studies was because of the importance of event similarity. In Table 1, we show results on the maternal morbidity dataset in Section 6.1 using a variety of different M's and tethering penalties. You are correct that the added outcomes provide more power.
> These results further underscore the importance of choosing surrogate events carefully based on prior (e.g., clinical) knowledge for similarity-based penalties to be effective. We hypothesize that including additional outcomes is more forgiving for feature learning improvement but is less beneficial for penalty-based approaches when they are not all closely related, as is the case in this example (shown by the pairwise comparisons in Figure 3). It may be possible to refine our strategy for settings with M > 2, and we intend to explore this in follow-up work.
> Ultimately, we believe that our theoretical and experimental insight into the importance of event similarity and event rate is an important contribution to the field even if it does focus on the case of M=2. However, we agree that expanding the discussion on the case of M > 2 would be an important addition to our paper, and we will add this in our final version.
>
> Table 1: The AUC of stroke prediction across 10 iterations using different outcome combinations with and without the CET penalty.
> | Method | Outcomes Used | CET Penalty | AUC Mean | AUC Std |
> |-|-|-|-|-|
> | NN | Stroke | N | 0.626 | 0.009 |
> | MLL-NN | Stroke + Hyp Crisis | N | 0.652 | 0.008 |
> | CET-NN | Stroke + Hyp Crisis | Y | 0.670 | 0.017 |
> | MLL-NN | All | N | 0.670 | 0.008 |
> | CET-NN | All | Y | 0.672 | 0.009 |
>
> - *Splitting strategy in Section 6.1*: We explained our reasoning for not following the standard splitting strategy for this task at the beginning of Section 6.1. Our primary objective was to closely emulate real-world medical datasets where the number of rare events is very small. Most real-world medical datasets are restricted in size due to high costs and data-sharing constraints. The maternal morbidity dataset we used was uniquely large, where even rare outcomes of interest had a substantial number of events due to the sheer number of samples. Therefore, we used only a portion of this dataset for training to emulate the more typical scenario. While, as you noted, allocating more data to the test set led to more consistent evaluation results, this was not our primary reason for this splitting strategy. We will expand on our motivation for this splitting strategy in the final version.

---

### Official Review · Reviewer_W1Nx · 2024-03-15

**Q2-1 Originality-Novelty:** 2
**Q2-2 Correctness-Technical Quality:** 3
**Q2-5 Clarity Of Writing:** 2

**Q1 Summary And Contributions:**

The paper is about the task of learning from observational data to predict rare medical events. The main idea is to exploit the concept of surrogate or related outcomes to the one of our main interest, which share etiology or underlying risk factors with the event of our interest. The paper builds on  regularized multi-label learning, bu developing two different variants. The main focus of the paper is on proving that the variants are effective to manage rare event prediction, The manuscripts first studies and derives the asymptotic properties of the two variants while also giving the theoretical insight into the convergence rates of the estimators on which the variants leverage. The results of a set of numerical experiments, based on synthetic data and on real world data, are summarized.

**Q2-3 Extent To Which Claims Are Supported By Evidence:**

2: Fair: the main claims are somewhat supported by evidence (but the experimental evaluation may be weak, or does not match entirely with the claims, important baselines may be missing, proofs contain important ideas but lack rigor, algorithmic details are only discussed superficially, references are imprecise, assumptions are not sufficiently motivated or explicated, etc.).

**Q2-4 Reproducibility:**

3: Good: key resources (e.g. proofs, code, data) are available and key details (e.g. proofs, experimental setup) are sufficiently well-described for competent researchers to confidently reproduce the main results.

**Q3 Main Strengths:**

1) the tackled problem is relevant
2) results of synthetic experiments confirms the claims
3) te state of the art is good

**Q4 Main Weakness:**

1) the paper is not easy to follow
2) results of numerical experiments ofn real world data tell a different story than that told by synthetic experiments, the improvement in AUC with respect to the baseline are extremely limited
3) the contibutions are limited and it is not given any indications on how to find surrogate events

**Q5 Detailed Comments To The Authors:**

I think the paper could read much better whether you better balance the maths and the explanation of it.
The results of numerical experiments on real world data do not fully convince me that the approach is that useful, or say it in different words, the improvement is marginal when compare to the baseline model.
I suggest to tackle more in detail how difficult and likely is to find surrogate events in general domain problems. Such a discussion could make a relevant contibution to spread your work.

**Q9 Complying With Reviewing Instructions:**

Yes

---

> ### Author Rebuttal · Authors · 2024-04-05
>
> We appreciate your detailed review and constructive feedback. In this response, we aim to address your concerns and provide additional clarification on the contributions of our paper. In particular, we discuss our *experimental results*, the topic of *finding surrogate events*, and the *clarity of the writing*.
>
> - *Experimental results*: We understand that the lack of consistent improvement our method has over the baseline in the various experimental results is a point of concern. However, we note that when a surrogate event is chosen well, as is the case when we use a hypertensive crisis to help predict stroke, our proposed method does have a significant improvement over both the single-label and multi-label learning baselines (the far left group of boxplots in Figure 3). Importantly, we also note that our experimental results show that our method does not hurt performance in the case that a more common surrogate event is not chosen perfectly and emphasize that even a small gain in performance is important when predicting life-altering medical events.
> A natural follow-up question is why we did not consider more real-world examples where the surrogate event is chosen well. In response to this, we note that while the problem characteristics motivating this work are common in healthcare settings, the corresponding datasets are difficult or expensive to obtain due to privacy and other concerns. Because of this, our experimental results repurposed existing datasets to fit our use case and were limited by the existing outcomes to show the potential gains and limitations of our method. Our results establish the potential benefit of building datasets that include more common events that are not of primary concern but associated with a rarer event of interest for improved prediction via ours and other methods.
>
> - *Finding surrogate events*: We note that we saw the most improvement in predicting a rare event (stroke) when we used a more common event that is known to be closely related physiologically and shares clinical risk factors (hypertensive crisis) (Pistoia et al, 2016). Whereas, our other real-world examples used rare and common event combinations without a similarly strong known physiological link. This finding suggests that previous literature may be the best resource for helping identify suitable surrogate events. For example, the various components of the atopic march often occurring in childhood, including asthma, eczema, and allergic rhinitis, share common underlying etiology and could be used to help predict one another using our approach (Hill \& Spergel, 2018).
> Overall, we agree that including more insight into how to choose a surrogate event will go a long way toward making our approach more accessible and useful for others. We appreciate your recommendation and will use the additional space in the final version of our paper to discuss this further.
>
> - *Clarity of writing*: We appreciate your recommendation of better balancing the math and explanation of it. We believe that we can address your concern by removing some of the notation from our textual explanations and focusing more on the implications of our theorems. For example, the most intuitive way to interpret the results from Theorem 4.2 is that for our approach to improve upon standard ridge regression, the parameter vector for the rare event of interest needs to be closer to the parameter vector for the more common surrogate event than it is to the zero vector. We will address this by adding more easy-to-follow explanations of our theorems to the main text and moving any notation-heavy explanations to the Supplementary Section where appropriate.
> As we continue to improve the clarity of our work, please let us know if there are additional specific portions that should be improved.
>
> **Citations:**
> - Hill, David A., and Jonathan M. Spergel. "The atopic march: critical evidence and clinical relevance." *Annals of Allergy, Asthma \& Immunology* 120.2 (2018): 131-137.
> - Pistoia, Francesca, et al. "Hypertension and stroke: epidemiological aspects and clinical evaluation." *High Blood Pressure \& Cardiovascular Prevention* 23 (2016): 9-18.

---

### Official Review · Reviewer_77q2 · 2024-03-22

**Q2-1 Originality-Novelty:** 2
**Q2-2 Correctness-Technical Quality:** 2
**Q2-5 Clarity Of Writing:** 2

**Q1 Summary And Contributions:**

As the title sounds it creates expectation to see a method about event prediction in longitudinal data, but the problem is multi class classification of rare events (categories) which involves severe imbalance. the authors introduce the problem in medicine and the need, and propose a method (common tethering with logistic regression, and another version with neural networks) for multi class. then the method is evaluated on simulations, and on real data, testing it with various parameters of the method, but not in comparison with alternative methods.

**Q2-3 Extent To Which Claims Are Supported By Evidence:**

2: Fair: the main claims are somewhat supported by evidence (but the experimental evaluation may be weak, or does not match entirely with the claims, important baselines may be missing, proofs contain important ideas but lack rigor, algorithmic details are only discussed superficially, references are imprecise, assumptions are not sufficiently motivated or explicated, etc.).

**Q2-4 Reproducibility:**

2: Fair: key resources (e.g. proofs, code, data) are unavailable but key details (e.g. proof sketches, experimental setup) are sufficiently well-described for an expert to confidently reproduce the main results.

**Q3 Main Strengths:**

the paper refers to an important problem
the paper proposes a emthod
there is an evaluation

**Q4 Main Weakness:**

the evaluation goals are not clearly stated and presented
the method is evaluated on only two datasets.
the method is not compared to alternative existing methods

**Q5 Detailed Comments To The Authors:**

The paper requires better structuring. In the evaluation start with stateing your research questions clearly. it will give context. then explain your experiments design so that they answer the questions. The evaluation must include some comparison to an existing alternative, as a baseline, to show the advantages of your proposed method. will be good to explain more the figures in the text. its not always clear.

**Q9 Complying With Reviewing Instructions:**

Yes

---

> ### Author Rebuttal · Authors · 2024-04-05
>
> We thank you for taking the time to review our paper and for your suggestions on how we can improve it. Our response is broken into separate topics to address each of your comments. We discuss *comparison to a baseline and alternative existing methods*, *number of real-world datasets*, and *evaluation goals*.
>
> - *Comparison to a baseline and alternative existing methods*: We compared our CET-LR to a single-label learning version of logistic regression and we compared our CET-NN to single-label learning and multi-label learning neural networks. We believe these methods are the ideal baselines because they are the same as our method except for the penalty we are proposing. Therefore the comparison allows us to isolate the effect of that penalty. This comparison approach his commonly used in ML literature that proposes a new penalty term and can be seen in the recent publication we cite by Faletto and Bien (2023).
> Thus, we believe a comparison to other methods is less helpful in terms of understanding the effect of our penalty term as separate from the many other factors affecting model performance. That being said, we do appreciate and agree that comparing to alternative methods will provide more context into how our approach performs compared to other SOTA approaches. We include results for three additional baselines -- a transfer learning approach, Firth’s logistic regression (Heinze \& Schemper, 2002; Doerken et al., 2019) and gradient-boosted trees -- on our maternal morbidity real-world data in Table 1, and will include similar results for our simulations and early autism prediction dataset in the final version.
>
> - *Number of real-world datasets*: Although additional results are always helpful, we respectfully disagree with your comment that “we only evaluated on two datasets” is a substantial weakness of our paper sufficient to warrant a lower score. The vast majority of ML papers that focus on medical applications consider only one (Pillai et al., 2023; Fatemi et al., 2022) or two (Wald \& Saria, 2023; Zhu et al., 2022; Liu \& Lin, 2023) datasets due to the difficulty of obtaining such data. Furthermore, we note that we had the added obstacle of requiring a dataset with a more common event closely related to a very rare event of interest. Ultimately, we were able to run on two real-world datasets that showed the potential of our method when a common event and rare event were chosen well - which was particularly the case when we used hypertensive crisis to help predict stroke (far left group of bars in Figure 3). We also supplemented these two real-world datasets with two distinct simulation settings under a range of conditions (e.g., event rate, event similarity) that help showcase both the potential as well as the limitations of our method on future datasets.
>
> - *Evaluation goals*: Our simulation studies aimed to determine three things: how (i) the underlying data generation process, (ii) the event similarity between the common and rare outcomes, and (iii) the event rate for the common outcome each impact the effectiveness of our proposed CET penalty. To address (i) we considered both a linear and non-linear data generation process for each of our experiments. To address (ii) we varied how similar the rare and common event outcomes were (Section 5.2). To address (iii) we varied to event rate of the more common outcome (Section 5.3).
> We outline these objectives and the steps we took to tackle each in the first paragraph of Section 5. However, we agree that this could be made clearer by restructuring the text and referencing the applicable subsections where we address each aim. We will make these changes in our final version.
> As for our real-world examples, we were concerned with assessing predictive performance using a variety of different common events to help predict either a stroke (Section 6.1) or the development of autism (Section 6.2). This section is a bit dense as we were limited by space. We will use the added space in our final version to describe the dataset in more detail and clarify the prediction tasks.
> Finally, we agree that more detailed explanations of the figures in both the descriptions and text will help improve clarity. We will be sure to add these in our final version.
>
>
> Table 1: The AUC of stroke prediction across 10 iterations with hypertensive crisis selected as surrogate event.
> | Category           | Method                 | AUC Mean | AUC Std |
> |-|-|-|-|
> | CET Methods | CET-LR | 0.638 | 0.012 |
> | | CET-NN | 0.670 | 0.017 |
> | Baseline Methods | LR | 0.605 | 0.018 |
> | | NN | 0.626 | 0.009 |
> | | Multilabel NN | 0.652 | 0.008 |
> | Comparison Methods | Firth LR* | 0.633 | 0.019 |
> | | Gradient Boosted Trees | 0.643 | 0.016 |
> | | Transfer Learning NN | 0.639 | 0.012 |
> *Note: For Firth’s LR we first ran feature selection similar to (Doerken et al., 2019), because Firth fails to converge on larger datasets with linearly dependent variables. We used LASSO for feature selection.

---

### Official Review · Reviewer_XNvS · 2024-03-22

**Q2-1 Originality-Novelty:** 3
**Q2-2 Correctness-Technical Quality:** 3
**Q2-5 Clarity Of Writing:** 4

**Q1 Summary And Contributions:**

This paper proposes a straightforward adaptation of existing penalty-based multi-task learning methods to rare event prediction. They present theoretical results for the bias and variance of this method and perform extensive empirical validation.

**Q2-3 Extent To Which Claims Are Supported By Evidence:**

4: Excellent: all claims are supported by very convincing evidence (in the form of comprehensive experimental evaluation, rigorous mathematical proofs, detailed (pseudo-)code, precise references, well-motivated and realistic assumptions) and the authors deliver what they promise.

**Q2-4 Reproducibility:**

4: Excellent: key resources (e.g. proofs, code, data) are available and key details (e.g. proof sketches, experimental setup) are comprehensively described for competent researchers to confidently and easily reproduce the main results.

**Q3 Main Strengths:**

I found this paper to be extremely well-written and, frankly, an enjoyable and educational read. The method is simple and does not deviate too far from existing methods, but it is applied thoughtfully to an absolutely critical problem in ML for healthcare. The theoretical results add to our understanding of this method and they are presented in a clear and intuitive way. The experiments are convincing and well described.

**Q4 Main Weakness:**

Minor weaknesses described below.

**Q5 Detailed Comments To The Authors:**

1. I think it is worth more clearly laying out the distinction between your proposal and Lapedriza et al. If I understood it correctly, Eq. 2 is equivalent to Lapedriza et al. in the case when M=2.
2. Relatedly, can you expand on what happens if M > 2? The advantage of the Lapedriza et al. approach is that the number of penalties increases linearly in M. Would you include penalties for all pairs of coefficients or just the coefficients for the target rare event? Was this tried in the preeclampsia data?
3. I think it is worth noting in the discussion some of the potential considerations and risks from this type of pooling. For example, what type of biases might be introduced? It is my understanding that ASD is more strongly associated with sex than ADHD. Complicating things, women and girls tend to be underdiagnosed to a greater degree than men and boys. Might this type of pooling lead to a sex bias in the resulting model?

**Q9 Complying With Reviewing Instructions:**

Yes

---

> ### Author Rebuttal · Authors · 2024-04-07
>
> We thank you for taking the time to leave a detailed and thoughtful review of our paper. We are glad you enjoyed reading it and will use this response to address the questions you raised in your comments. We organize our response into three sections addressing the *relation to Lapedriza et al.*, the *case of M > 2*, and the *considerations and risks of pooling outcomes*.
>
> - *Relation to Lapedriza et al.*: You are correct that Equation 2 is equivalent Lapedriza et al. in the case when M=2. We will be sure to state this more clearly in our final version.
> Our work expands on Lapedriza et al. in two important ways. It is the first to explore via both simulation and theory, the impact of event relatedness and event rate when using shrinkage penalties like those proposed by Lapedriza et al. and our method. Secondly, we incorporate feature learning via a NN architecture to allow our approach to extend to more complicated non-linear setups. We summarize this distinction at the end of our Related Works section, but agree with you that it is worthwhile to more clearly lay out this distinction. We thank you for this suggestion and will incorporate it into our final version.
> - *Case of M > 2*: The baseline approach for M>2 incorporates M choose 2 penalty terms and thus does not scale linearly. However, in light of our findings, we advocate for targeted selection of a limited number of outcomes with related clinical etiology or risk factors, therefore we believe this is not a limitation in practice. Furthermore, with domain knowledge the framework can be modified to only incorporate penalty terms between the rare event and a select number of the most similar common events.
> We did explore using all 4 outcomes to help predict stroke in the maternal morbidity dataset by including a penalty for all pairs of coefficients. In this setup, we saw negligible improvement in the AUC (went from 0.670 without CET penalty to 0.672 with the CET penalty). We hypothesize that including additional outcomes is more forgiving for feature learning improvement but is less beneficial for similarity-based penalty approaches when all the outcomes are not closely related, as is the case in this example (as shown by the pairwise comparisons in Figure 3). It may be possible to refine our strategy for settings with M > 2, and we intend to explore this in follow-up work.
> In our autism dataset, the common outcome was a composite of several neurodevelopmental diagnoses, including language and motor delays, that can be separated to further explore this setting. We will include results on this dataset after separating these outcomes as well as results from the simulation setups and the maternal morbidity dataset in the final version of the paper.
> Ultimately, we chose to focus on the case where M=2 in our theory and simulation studies to simplify the exploration of how event similarity and event rate impact the effectiveness of our proposed approach. As we noted above, this was an important aspect of our paper's contribution as it has not been done in previous literature. However, we agree that expanding discussion on how to approach the case of M > 2 would be an important addition to our paper, and we will add this in our final version.
> - *Considerations and risks of pooling outcomes*: This is an excellent point that we had not considered and is highly relevant to our work. We will add a brief discussion of this point to the final version of our paper. As you said, both autism and ADHD are more common in boys, but the imbalance is greater for autism, and girls tend to be diagnosed less often and at a later age (see e.g. Loomes et al., 2017). Your point suggests a very interesting future direction to explore relationships between algorithmic bias/fairness and multilabel learning that we'll discuss in the final version and would like to explore in future work (if OK with you!).
> Along the same lines, racial biases could be exacerbated by using more common outcomes that are under or over-diagnosed in certain underrepresented racial groups. Indeed, the naive use of a multilabel approach could worsen existing biases by propagating bias from one (biased) outcome to another (less biased) outcome. For example, hypertensive-related adverse events are more common in non-Hispanic Black persons (Abrahamowicz et al., 2023), but the degree of imbalance differs between specific outcomes. Therefore, if race or highly correlated covariates are used as predictors, this penalty approach may result in miscalibrated predictions in the Black subpopulation. The ethical implications of this would depend on the details and the application, but in general, there is potential for harm. Ultimately, we believe that a thoughtful discussion on these potential biases is an important conversation and one that we will add.

---

### Official Review · Reviewer_uBgP · 2024-03-24

**Q2-1 Originality-Novelty:** 3
**Q2-2 Correctness-Technical Quality:** 3
**Q2-5 Clarity Of Writing:** 4

**Q1 Summary And Contributions:**

The paper focuses on predicting rare clinical event by leveraging some more common and related events. The authors propose a variant of regularized multi-label learning (MML) method, called common event tethering (CET), and conduct a theoretical analysis on the asymptotic properties on CET's logistic regression variant. They also draw insights into the convergence rate of CET.

Furthermore, the authors set up simulations to show CET outperform single-label learning in terms of Spearman's rank correlation $(\rho)$ when event similarity (between rare and common) exceeds a certain threshold (e.g., 40%) and the performance continues to improve as the similarity grows. They also evaluate CET on two real-world medical datasets (to predict stroke and early autism respectively) in terms of AUC, and demonstrate CET outperform single-label training.

**Q2-3 Extent To Which Claims Are Supported By Evidence:**

4: Excellent: all claims are supported by very convincing evidence (in the form of comprehensive experimental evaluation, rigorous mathematical proofs, detailed (pseudo-)code, precise references, well-motivated and realistic assumptions) and the authors deliver what they promise.

**Q2-4 Reproducibility:**

4: Excellent: key resources (e.g. proofs, code, data) are available and key details (e.g. proof sketches, experimental setup) are comprehensively described for competent researchers to confidently and easily reproduce the main results.

**Q3 Main Strengths:**

- The motivation of CET seems intuitive and easy to understand, and the authors endeavor to rigorously draw theoretical evidences for CET.
- The proposed estimator is not only backed by simulations, but also experiments on real-world medical datasets.

**Q4 Main Weakness:**

- Multi-label learning doesn't always come to mind naturally when having an extreme data imbalance problem. It would be better to briefly elaborate why other techniques are not in the scope of this paper.

**Q5 Detailed Comments To The Authors:**

The proposed CET, at a high level, pulls estimated coefficients of $\theta_1$ on a rare event toward the estimated coefficients of $\theta_2$ on a more common event, in order to share information between the learned estimators. This is particularly useful when features of rare events cannot provide enough supervisory signals to the estimator, but complementary with more common events with potentially "overlapped" features the gap is shorten.

When I have two classes where one is rare and the other more common, there are several others machine learning setups besides multi-label learning. Transfer learning for a trained estimator on a more common event and fine-tuning on a rare common event is also a type of information share, although being one-way. Neural Networks with a shared encoder/layer and two different output classifiers also enables feature sharing. If the authors provide some context of why they are not in the scope, that would be great.

**Questions**
-  In Figure 3, the results aligns with clinical explanations in terms of how two rare events might share information. If not CET-NN but CET-LR is used, do the authors think CET-LR can be used in biomarker discovery?

**Minor Issues**
- In Section 3.1, should the parameter vector $\boldsymbol{\theta}=\left[\boldsymbol{\theta}_1, \boldsymbol{\theta}_2\right] \in \mathbb{R}^{2 p}$ or $\in \mathbb{R}^{2 d}$? Does it apply to the input features or the transformed features?

**Q9 Complying With Reviewing Instructions:**

Yes

---

> ### Author Rebuttal · Authors · 2024-04-07
>
> We thank you for your time and detailed review. We would like to use this response to address your comments and provide additional clarification. We have organized our response into sections on *other machine learning approaches*, *biomarker discovery and feature selection*, and *Section 3.1 parameter vector clarification* to clearly cover each of your questions.
>
> - *Other machine learning approaches*: You are right that there are other techniques available that can be used in setups like ours. The majority of well-established methods for handling imbalanced data are developed for the rare event itself, and lack a specific framework for integrating more common events. That being said, comparing with other single-learning baselines is a good idea to provide evidence that sharing information can be helpful. We will add a brief discussion on single-label learning approaches to our review. And in this response, we include comparisons of our method with Firth logistic regression and gradient boosted trees. These methods have been shown to perform well in applications of rare event prediction with high-dimensional datasets similar to our setup (Doerken, 2019).
> We also agree with and appreciate your suggestion to compare with alternative methods for information sharing. We do review both multi-label and multi-task learning, but agree that also discussing transfer learning would be beneficial to position our approach among that line of work as well. We will include transfer learning in the discussion of related methods for our final version. We also want to point out that the neural network with a shared encoder and different output classifiers is a multi-label learning technique that corresponds to the CET-NN approach with no similarity penalty. Our experiments do compare to this technique (it is represented by green boxes in figures).
> In Table 1, we show the results for three additional baselines that include a transfer learning approach, Firth's logistic regression, and gradient boosted trees on our maternal morbidity dataset for stroke prediction. We will include similar results for our simulation study and autism prediction dataset in the final version of our paper.
>
> Table 1: The AUC of stroke prediction across 10 iterations with hypertensive crisis selected as surrogate event.
> | Category | Method | AUC Mean | AUC Std |
> |-|-|-|-|
> | CET Methods | CET-LR | 0.638 | 0.012 |
> | | CET-NN | 0.670 | 0.017 |
> | Baseline Methods | LR | 0.605 | 0.018 |
> | | NN | 0.626 | 0.009 |
> | | Multilabel NN | 0.652 | 0.008 |
> | Comparison Methods | Firth LR* | 0.633 | 0.019 |
> | | Gradient Boosted Trees | 0.643 | 0.016 |
> | | Transfer Learning NN | 0.639 | 0.012 |
> *Note: For Firth’s LR we first ran feature selection similar to (Doerken, 2019), because Firth fails to converge on larger datasets with linearly dependent variables. We used LASSO for feature selection.
>
> - *Biomarker discovery and feature selection*: We thank you for raising this practical question which helps us expand the potential applications of our approach. While we are not experts in biomarker discovery, variants of fused Lasso and multitask learning have been shown to be very successful at biomarker discovery (Zhou et al., 2012; Liang et al., 2023). In turn, we think that this would be a very promising application of our approach. If you are involved in biomarker discovery we would love to have you try it and hear how it works!
> More broadly than biomarker discovery, your question made us think of potential applications of CET-LR for variable selection when an L1 version of the CET penalty is used. While exploring these additional applications may be outside the scope of this paper, we will include these ideas in our discussion for future work and possible applications. Thank you again for this thoughtful question as it has gotten us thinking of lots of potential applications to pursue going forward with this method!
>
> - *Section 3.1 parameter vector clarification*: In Section 3.1, the $\boldsymbol{\theta}\in\mathbb{R}^{2p}$ as it applies to the input features. Upon rereading this section, we see how our framing of $h(\mathbf{x}_i) = \mathbf{H}\mathbf{x}_i$ where $\mathbf{H}\in\mathbb{R}^{d\times p}$ makes this confusing. We wanted to portray that as long as the true underlying risk factors are *linear* combinations of the covariates then a logistic regression model is correctly specified. However, we agree that the notation leads to more confusion than insight here. We will remove the notion of the matrix $\mathbf{H}$ and will just say that as long as $h(\mathbf{x}_i)$ is linear then logistic regression is correctly specified.

---

### Meta-Review · Area_Chair_apTy · 2024-04-17

While the reviewers did not reach a consensus, there are several strengths of the paper identified by multiple reviewers, including the quality of both the theoretical and experimental results. Reviewers are more mixed about the quality of presentation and novelty. The authors have addressed some of the reviewers' concerns in their rebuttal. Furthermore, I find the experimental evaluation on two medical application settings to be appropriate and do not view it as a negative. I believe that this work would interest much of the UAI audience.